# CRISP: Curriculum based Sequential neural decoders for Polar code family

## Abstract

Polar codes are widely used state-of-the-art codes for reliable communication that have recently been included in the $5^{\text{th}}$ generation wireless standards (5G). However, there remains room for the design of polar decoders that are both efficient and reliable in the short blocklength regime. Motivated by recent successes of data-driven channel decoders, we introduce a novel **CurRI**culum based **S**equential neural decoder for **P**olar codes (CRISP)[1]. We design a principled curriculum, guided by information-theoretic insights, to train CRISP and show that it outperforms the successive-cancellation (SC) decoder and attains near-optimal reliability performance on the Polar$(32, 16)$ and Polar$(64, 22)$ codes. The choice of the proposed curriculum is critical in achieving the accuracy gains of CRISP, as we show by comparing against other curricula. More notably, CRISP can be readily extended to Polarization-Adjusted-Convolutional (PAC) codes, where existing SC decoders are significantly less reliable. To the best of our knowledge, CRISP constructs the first data-driven decoder for PAC codes and attains near-optimal performance on the PAC$(32, 16)$ code.

## 1 Introduction

Error-correcting codes (codes) are the backbone of modern digital communication. Codes, composed of (encoder, decoder) pairs, ensure reliable data transmission even under noisy conditions. Since the groundbreaking work of Shannon (1948), several landmark codes have been proposed: Convolutional codes, low-density parity-check (LDPC) codes, Turbo codes, Polar codes, and more recently, Polarization-Adjusted-Convolutional (PAC) codes (Richardson & Urbanke, 2008). In particular, polar codes, introduced by Arıkan (2009), are widely used in practice owing to their reliable performance in the short blocklength regime. A family of variants of polar codes known as PAC codes further improves performance, nearly achieving the fundamental lower bound on the performance of any code at finite lengths, albeit at a higher decoding complexity (Arıkan, 2019). In this paper, we focus on the *decoding* of these two classes of codes, jointly termed the "Polar code family".

The polar family exhibits several crucial information-theoretic properties; practical finite-length performance, however, depends on high complexity decoders. This search for the design of efficient and reliable decoders for the Polar family is the focus of substantial research in the past decade. **(a) Polar codes:** The classical successive cancellation (SC) decoder achieves information-theoretic capacity asymptotically, but performs poorly at finite blocklengths compared to the optimal maximum a posteriori (MAP) decoder (Arıkan, 2019). To improve upon the reliability of SC, several polar decoders have been proposed in the literature (Sec. 6). One such notable result is the celebrated Successive-Cancellation-with-List (SCL) decoder (Tal & Vardy, 2015). SCL improves upon the reliability of SC and approaches that of the MAP with increasing list size (and complexity). **(b) PAC codes:** The sequential "Fano decoder" (Fano, 1963) allows PAC codes to perform information-theoretically near-optimally; however, the decoding time is long and variable (Rowshan et al., 2020a). Although SC is efficient, $O(n \log n)$, its performance with PAC codes is significantly worse than that of the Fano decoder. Several works (Yao et al., 2021; Rowshan et al., 2020b; Zhu et al., 2020; Rowshan & Viterbo, 2021b;a; Sun et al., 2021) propose ameliorations; it is safe to say that constructing efficient and reliable decoders for the Polar family is an active area of research and of

---

[1]Source code available at the following link.

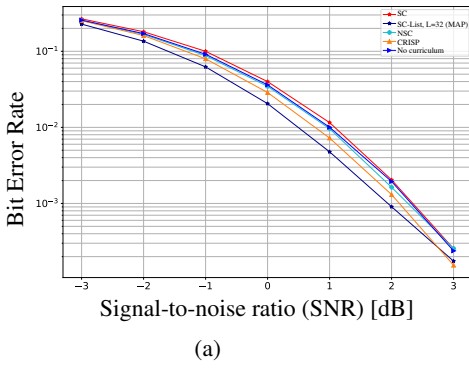 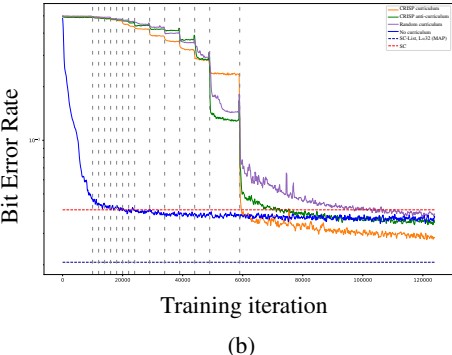

(a)                                                    (b)

Figure 1: (a) CRISP achieves near-MAP reliability for Polar$(64, 22)$ code on the AWGN channel. (b) Our proposed curriculum is crucial for the gains CRISP attains over the baselines; details in Sec. 4.

utmost practical interest given the advent of Polar codes in 5G wireless cellular standards. The design of efficient and reliable decoders for the Polar family is the focus of this paper.

In this paper, we introduce a novel **CurRI**culum based **S**equential neural decoder for **P**olar code family (CRISP). When the proposed curriculum is applied to neural network decoder training, thus trained decoders outperform existing baselines and attain near-MAP reliabilty on Polar$(64, 22)$, Polar$(32, 16)$ and PAC$(32, 16)$ codes while maintaining low computational complexity (Figs. 1, 5, Table 1). CRISP builds upon an inherent nested hierarchy of polar codes; a Polar$(n, k)$ code subsumes all the codewords of lower-rate subcodes Polar$(n, i), 1 \le i \le k$ (Sec. 2.2). We provide principled curriculum of training on examples from a sequence of sub-codes along this hierarchy, and demonstrate that the proposed curriculum is critical in attaining near-optimal performance (Sec. 4).

Curriculum-learning (CL) is a training strategy to train machine learning models, starting with easier subtasks and then gradually increasing the difficulty of the tasks (Wang et al., 2021). (Elman, 1993), a seminal work, was one of the first to employ CL for supervised tasks, highlighting the importance of "starting small". Later, Bengio et al. (2009) formalized the notion of CL and studied when and why CL helps in the context of visual and language learning (Wu et al., 2020; Wang et al., 2021). In recent years, many empirical studies have shown that CL improves generalization and convergence rate of various models in domains such as computer vision (Pentina et al., 2015; Jesson et al., 2017; Morerio et al., 2017; Guo et al., 2018; Wang et al., 2019), natural language processing (Cirik et al., 2016; Platanios et al., 2019), speech processing (Amodei et al., 2016; Gao et al., 2016; 2018), generative modeling (Karras et al., 2017; Wang et al., 2018), and neural program generation (Zaremba & Sutskever, 2014; Reed & De Freitas, 2015). Viewed from this context, our results add decoding of algebraic codes (of the Polar family) to the domain of successes of supervised CL. In summary, we make the following contributions:

- We introduce CRISP, a novel curriculum-based sequential neural decoder for the Polar code family. Guided by information-theoretic insights, we propose CL-based techniques to train CRISP, that are crucial for its superior performance (Sec. 3).

- We demonstrate that CRISP attains near-optimal reliability performance on Polar$(64, 22)$ and Polar$(32, 16)$ codes whilst achieving a good throughput (Sec. 4.1 and Sec. 4.2).

- Compared to Fano's decoder, the CRISP decoder has significantly higher throughput and attains near-MAP reliability for the PAC$(32, 16)$ code. To the best of our knowledge, this is the first learning-based PAC decoder to achieve this performance (Sec. 4.4).

## 2 PROBLEM FORMULATION

In this section we formally define the channel decoding problem and provide background on the Polar code family. Our notation is the following: we denote Euclidean vectors by small bold face letters

$\boldsymbol{x}, \boldsymbol{y}$, etc. $[n] \triangleq \{1, 2, \ldots, n\}$. For $\boldsymbol{m} \in \mathbb{R}^n$, $\boldsymbol{m}_{<i} \triangleq (m_1, \ldots, m_{i-1})$. $\mathcal{N}(0, \boldsymbol{I}_n)$ denotes a standard Gaussian distribution in $\mathbb{R}^n$. $\boldsymbol{u} \oplus \boldsymbol{v}$ denotes the bitwise XOR of two binary vectors $\boldsymbol{u}, \boldsymbol{v} \in \{0, 1\}^\ell$.

## 2.1 CHANNEL DECODING

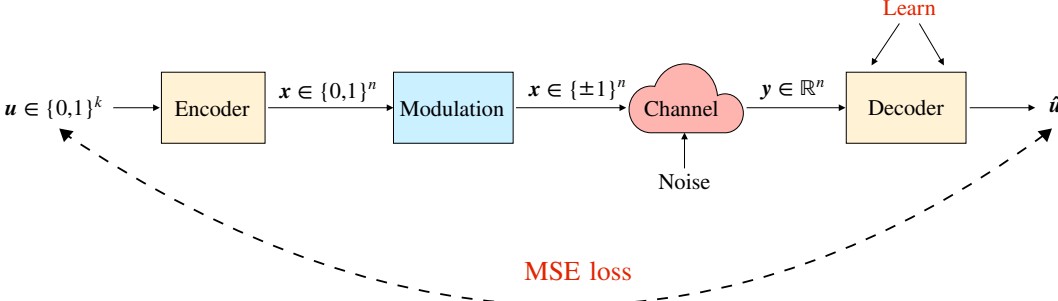

Figure 2: Channel decoding problem.

The primary goal of channel decoding is to design efficient decoders that can correctly recover the message bits upon receiving codewords corrupted by noise (Fig. 2). More precisely, let $\boldsymbol{u} = (u_1, \ldots, u_k) \in \{0, 1\}^k$ denote a block of *information/message* bits that we wish to transmit. An encoder $g : \{0, 1\}^k \to \{0, 1\}^n$ maps these message bits into a binary codeword $\boldsymbol{x}$ of length $n$, i.e. $\boldsymbol{x} = g(\boldsymbol{u})$. The encoded bits $\boldsymbol{x}$ are modulated via Binary Phase Shift Keying (BPSK), i.e. $\boldsymbol{x} \mapsto 1 - 2\boldsymbol{x} \in \{\pm 1\}^n$, and are transmitted across the channel. We denote both the modulated and unmodulated codewords as $\boldsymbol{x}$. The channel, denoted as $P_{Y|X}(\cdot|\cdot)$, corrupts the codeword $\boldsymbol{x}$ to its noisy version $\boldsymbol{y} \in \mathbb{R}^n$. Upon receiving the corrupted codeword, the decoder $f_\theta$ estimates the message bits as $\hat{\boldsymbol{u}} = f_\theta(\boldsymbol{y})$. The performance of the decoder is measured using standard error metrics such as Bit-Error-Rate (BER) or Block-Error-Rate (BLER): $\mathrm{BER}(f_\theta) \triangleq (1/k) \sum_i \mathbb{P}[\hat{\boldsymbol{u}}_i \neq \boldsymbol{u}_i]$, whereas $\mathrm{BLER}(f_\theta) \triangleq \mathbb{P}[\hat{\boldsymbol{u}} \neq \boldsymbol{u}]$.

Given an encoder $g$ with code parameters $(n, k)$ and a channel $P_{Y|X}$, the channel decoding problem can be mathematically formulated as:

$$\theta \in \arg\min_\theta \mathrm{BER}(f_\theta), \qquad (1)$$

which is a joint classification of $k$ binary classes. To train the parameters $\theta$, we use the mean-square-error (MSE) loss as a differentiable surrogate to the objective in Eq. 1. It is well known in the literature that naively parametrizing $f_\theta$ by general-purpose neural networks does not work well and they perform poorly even for small blocklengths like $n = 16$ (Gruber et al., 2017). Hence it is essential to use efficient decoding architectures that capitalize on the structure of the encoder $g$ (Kim et al., 2018b; Chen & Ye, 2021). To this end, we focus on a popular class of codes, the *Polar code family*, that comprises two state-of-the-art codes: Polar codes (Arikan, 2009) and Polarization-Adjusted-Convolutional (PAC) codes (Arıkan, 2019). Both these codes are closely related and hence we first focus on polar codes in Sec. 2.2. In Sec. 3, we present CRISP, our novel curriculum-learning based neural decoder to decode polar codes. In Sec. 4.4 we detail PAC codes.

## 2.2 POLAR CODES

**Encoding.** Polar codes, introduced in Arikan (2009), were the first codes to be theoretically proven to achieve capacity for any binary-input discrete memoryless channel. Their encoding is defined as follows: let $(n, k)$ be the code parameters with $n = 2^p, 1 \leq k \leq n$. In order to encode a block of message bits $\boldsymbol{u} = (u_1, \ldots, u_k) \in \{0, 1\}^k$, we first embed them into a source message vector $\boldsymbol{m} \triangleq (m_1, \ldots, m_n) = (0, \ldots, u_1, 0, \ldots, u_2, 0, \ldots, u_k, 0, \ldots) \in \{0, 1\}^n$, where $\boldsymbol{m}_{I_k} = \boldsymbol{u}$ and $\boldsymbol{m}_{I_k^C} = 0$ for some $I_k \subseteq [n]$. Since the message block $\boldsymbol{m}$ contains the information bits $\boldsymbol{u}$ only at the indices pertaining to $I_k$, the set $I_k$ is called the *information set*, and its complement $I_k^C$ the *frozen set*. For the set $I_k$, we first compute the capacities of the $n$ individual polar bit channels and rank them in their increasing order (Tal & Vardy, 2013). Then $I_k$ picks the top $k$ out of them. For example, Polar(4, 2) has the ordering $m_1 < m_2 = m_3 < m_4$ and $I_k = \{2, 4\}$, and thus $\boldsymbol{m} = (0, m_2, 0, m_4)$. Similarly, Polar(8, 4) has $m_1 < m_2 < m_3 < m_5 < m_4 < m_6 < m_7 < m_8$, $I_4 = \{4, 6, 7, 8\}$ and $\boldsymbol{m} = (0, 0, 0, m_4, 0, m_6, m_7, m_8)$.

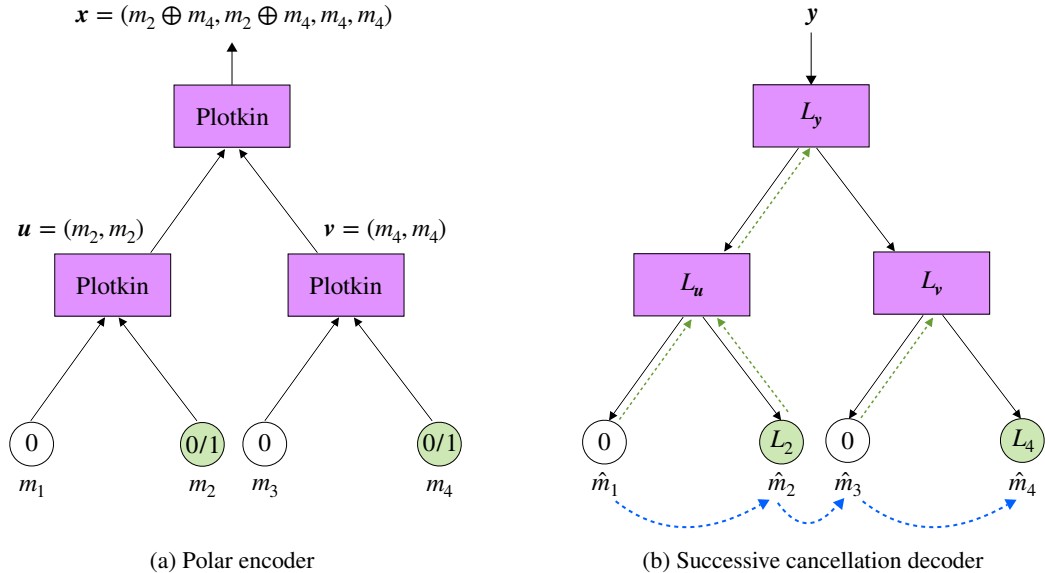

Figure 3: Polar$(4, 2)$: (a) Polar encoding via Plotkin tree; (b) Blue arrows indicate the decoding order.

Finally, we obtain the polar codeword $\boldsymbol{x}$ = $\mathrm{PlotkinTree}(\boldsymbol{m})$, where the mapping $\mathrm{PlotkinTree} : \{0,1\}^n \to \{0,1\}^n$ is given by a complete binary tree, known as Plotkin tree (Plotkin, 1960). Fig. 3a details the Plotkin tree for Polar$(4, 2)$. Plotkin tree takes the input message block $\boldsymbol{m} \in \{0,1\}^n$ at the leaves and applies the "Plotkin" function at each of its internal nodes recursively to obtain the codeword $\boldsymbol{x} \in \{0,1\}^n$ at the root. The function $\mathrm{Plotkin} : \{0,1\}^\ell \times \{0,1\}^\ell \to \{0,1\}^{2\ell}, \ell \in \mathbb{N}$, is defined as

$$\mathrm{Plotkin}(\boldsymbol{u}, \boldsymbol{v}) \triangleq (\boldsymbol{u} \oplus \boldsymbol{v}, \boldsymbol{v}).$$

For example, in Fig. 3a, starting with the message block $\boldsymbol{m} = (0, m_2, 0, m_4)$ at the leaves, we first obtain $\boldsymbol{u} = \mathrm{Plotkin}(0, m_2) = (m_2, m_2)$ and $\boldsymbol{v} = \mathrm{Plotkin}(0, m_4) = (m_4, m_4)$. Applying the function once more, we obtain the codeword $\boldsymbol{x} = \mathrm{Plotkin}(\boldsymbol{u}, \boldsymbol{v}) = (m_2 \oplus m_4, m_2 \oplus m_4, m_4, m_4)$.

**Decoding.** The successive-cancellation (SC) algorithm is one of the most efficient decoders for polar codes, with a decoding complexity of $O(n \log n)$. The basic principle behind the SC algorithm is to sequentially decode one message bit $m_i$ at a time according to the conditional log-likelihood ratio (LLR), $L_i \triangleq \log(\mathbb{P}[m_i = 0|\boldsymbol{y}, \hat{\boldsymbol{m}}_{<i}]/\mathbb{P}[m_i = 1|\boldsymbol{y}, \hat{\boldsymbol{m}}_{<i}])$, given the corrupted codeword $\boldsymbol{y}$ and previous decoded bits $\hat{\boldsymbol{m}}_{<i}$ for $i \in I_k$. Fig. 3b illustrates this for the Polar$(4, 2)$ code: for both the message bits $m_2$ and $m_4$, we compute these conditional LLRs and decode them via $\hat{m}_2 = \mathbb{1}\{L_2 < 0\}$ and $\hat{m}_4 = \mathbb{1}\{L_4 < 0\}$. Given the Plotkin tree structure, these LLRs can be efficiently computed sequentially using a depth-first-search based algorithm (App. A).

As discussed in Sec. 1, SC achieves the theoretically optimal performance only asymptotically, and its reliability is sub-optimal at finite blocklengths. SC-list (SCL) decoding improves upon its error-correction performance by maintaining a list of $L$ candidate paths at any time step and choosing the best among them in the end. In fact, for a reasonably large list size $L$, SCL achieves MAP performance at the cost of increased complexity $O(Ln \log n)$, as highlighted in Table 1.

## 3 CRISP: CURRICULUM BASED SEQUENTIAL NEURAL DECODER FOR POLAR FAMILY

We design CRISP, a curriculum-learning-based sequential neural decoder for polar codes that strictly outperforms the SC algorithm and existing baselines. CRISP uses a sequential RNN decoder, powered by gated recurrent units (GRU) (Chung et al., 2014), to decode one bit at a time. Instead of standard training techniques, we design a novel curriculum, guided by information-theoretic insights, to train the RNN to learn good decoders. Fig. 4 illustrates our approach.

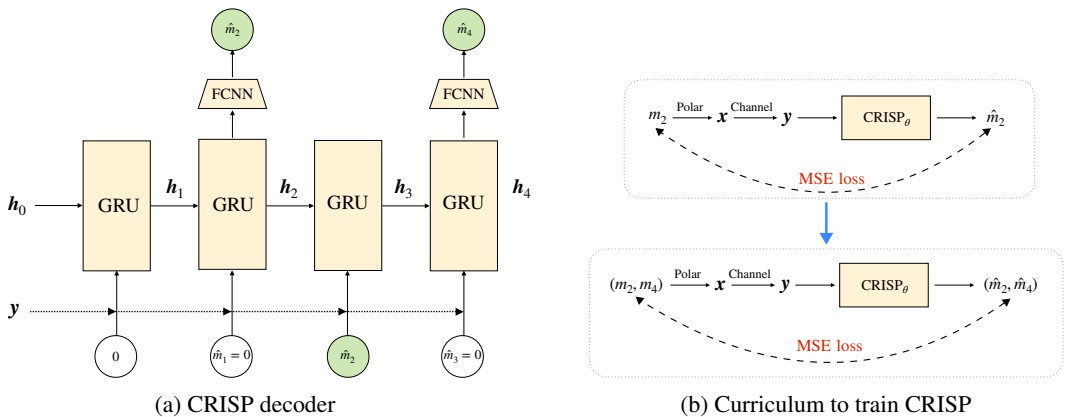

(a) CRISP decoder                                    (b) Curriculum to train CRISP

Figure 4: CRISP decoder and its training by curriculum-learning for Polar$(2, 4)$.

**CRISP decoder.** We use the Polar$(4, 2)$ code as a guiding example to illustrate our CRISP decoder (Fig. 4a). This code has two message bits $(m_2, m_4)$ and the message block is $\boldsymbol{m} = (0, m_2, 0, m_4)$. Upon encoding it to the polar codeword $\boldsymbol{x} \in \{\pm1\}^4$ and receiving its noisy version $\boldsymbol{y} \in \mathbb{R}^4$, the decoder estimates the message as $\hat{\boldsymbol{m}} = (0, \hat{m}_2, 0, \hat{m}_4)$. Similar to SC, CRISP uses the sequential paradigm of decoding one bit $\hat{m}_i$ at a time by capitalizing on the previous decoded bits $\hat{\boldsymbol{m}}_{<i}$ and $\boldsymbol{y}$. To that end, we parametrize the bit estimate $\hat{m}_i$ conditioned on the past as a fully connected neural network (FCNN) that takes the hidden state $\boldsymbol{h}_i$ as its input. Here $\boldsymbol{h}_i$ denotes the hidden state of the GRU that implicitly encodes this past information $(\hat{\boldsymbol{m}}_{<i}, \boldsymbol{y})$ via GRU's recurrence equation, i.e.

$$\boldsymbol{h}_i = \text{GRU}_\theta(\boldsymbol{h}_{i-1}, \hat{m}_{i-1}, \boldsymbol{y}), \quad i \in \{1, 2, 3, 4\}, \tag{2}$$

$$\hat{m}_i | \boldsymbol{y}, \hat{\boldsymbol{m}}_{<i} = \text{FCNN}_\theta(\boldsymbol{h}_i), \quad i \in \{4, 2\}, \tag{3}$$

where $\theta$ denotes the FCNN and GRU parameters jointly. Henceforth we refer to our decoder as either CRISP or CRISP$_\theta$. Note that while the RNN is unrolled for $n = 4$ time steps (Eq. 2), we only estimate bits at $k = 2$ information indices, i.e. $\hat{m}_2$ and $\hat{m}_4$ (Eq. 3). A key drawback of SC is that a bit error at a position $i$ can contribute to the future bit errors $(> i)$, and it does not have a feedback mechanism to correct these error events. On the other hand, owing to the RNN's recurrence relation (Eq. 2), CRISP can learn to correct these mistakes through the gradient it receives (via backpropagation through time) during training.

**Curriculum-training of CRISP.** Given the decoding architecture of CRISP in Fig. 4a, a natural approach to train its parameters via supervised learning is to use a joint MSE loss function for both the bits $(\hat{m}_2, \hat{m}_4)$: $\text{MSE}(\hat{m}_2, \hat{m}_4) = (\hat{m}_2(\theta) - m_2)^2 + (\hat{m}_4(\theta) - m_4)^2$. However, as we highlight in Sec. 4.1 such an approach learns to fail better decoders than SC and gets stuck at local minima. To address this issue, we propose a curriculum-learning based approach to train the RNN parameters.

The key idea behind our curriculum training of CRISP is to decompose the problem of joint estimation of bits $(\hat{m}_2, \hat{m}_4)$ into a sequence of sub-problems with increasing difficulty: start with learning to estimate only the first bit $(\hat{m}_2)$ and progressively add one new message bit at each curriculum step $(\hat{m}_4)$ until we estimate the full message block $\boldsymbol{m} = (\hat{m}_2, \hat{m}_4)$. We freeze all the non-trainable message bits to zero during any curriculum step. In other words, in the first step, we freeze the bit $m_4$ and train the decoder only to estimate the bit $\hat{m}_2$ (i.e. the subcode corresponding to $k = 1$):

$$(m_2, m_4 = 0) \rightarrow \boldsymbol{m} = (0, m_2, 0, 0) \xrightarrow{\text{Polar}} \boldsymbol{x} \xrightarrow{\text{Channel}} \boldsymbol{y} \xrightarrow{\text{CRISP}_\theta} \hat{m}_2. \tag{4}$$

We use this trained $\theta$ as an initialization for the next task of estimating both the bits $(\hat{m}_2, \hat{m}_4)$:

$$(m_2, m_4) \rightarrow \boldsymbol{m} = (0, m_2, 0, m_4) \xrightarrow{\text{Polar}} \boldsymbol{x} \xrightarrow{\text{Channel}} \boldsymbol{y} \xrightarrow{\text{CRISP}_\theta} (\hat{m}_2, \hat{m}_4). \tag{5}$$

Fig. 4b illustrates this curriculum-learning approach. We note that the knowledge of decoding $\hat{m}_2$ when $m_4 = 0$ (Eq. 4) serves as a good initialization when we learn to decode $\hat{m}_2$ for a general $m_4 \in \{0, 1\}$ (Eq. 5). With such a curriculum aided training, we show in Sec. 4.1 (Figs. 1, 5) that the

CRISP decoder outperforms the existing baselines and attains near-optimal performance for a variety of blocklengths and codes. We interpret this in Sec. 4.3. We defer the training details to App. E.

**Left-to-Right (L2R) curriculum for Polar**$(n, k)$. For a general Polar$(n, k)$ code, we follow a similar curriculum to train CRISP$_\theta$. Denoting the index set by $I_k = \{i_1, i_2, \ldots, i_k\} \subseteq [n]$ in the increasing order of indices $i_1 < i_2 < \ldots < i_k$, our curriculum is given by: Train $\theta$ on $\hat{m}_{i_1} \to$ Train $\theta$ on $(\hat{m}_{i_1}, \hat{m}_{i_2}) \to \ldots \to$ Train $\theta$ on $(\hat{m}_{i_1}, \ldots, \hat{m}_{i_k})$. We term this curriculum *Left-to-Right (L2R)*. The anti-curriculum *R2L* refers to progressively training in the decreasing order of the indices in $I_k$.

## 4 MAIN RESULTS

In this section, we present numerical results for the CRISP decoder on the Polar code family.

### 4.1 AWGN CHANNEL

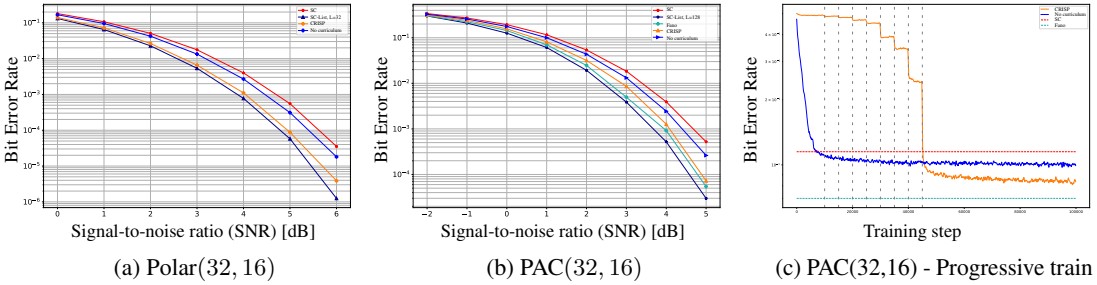

(a) Polar$(32, 16)$      (b) PAC$(32, 16)$      (c) PAC$(32,16)$ - Progressive train

Figure 5: CRISP outperforms baselines and attains near-MAP performance for Polar$(32, 16)$ and PAC$(32, 16)$ codes on the AWGN channel.

**Data generation**. The input message $\boldsymbol{u} \in \{0, 1\}^k$ is randomly drawn uniformly from the boolean hypercube and encoded as a polar codeword $\boldsymbol{x} \in \{\pm 1\}^n$. The classical additive white Gaussian noise (AWGN) channel, $\boldsymbol{y} = \boldsymbol{x} + \boldsymbol{z}, \boldsymbol{z} \sim \mathcal{N}(0, \sigma^2 \boldsymbol{I}_n)$, generates the training/test data $(\boldsymbol{y}, \boldsymbol{u})$ for the decoder. The signal-to-noise ratio, i.e. SNR $= -10 \log_{10} \sigma^2$, characterizes the noise level in the channel. Here we fix the channel to be AWGN in all our experiments, as per the standard convention (Kim et al., 2018b), and refer to App. D for additional results on fading and t-distributed channels. App. E details the training procedure. Once trained, we use the CRISP models for comparison against the baselines.

**Baselines**. The optimal channel decoder is the MAP estimator: $\hat{\boldsymbol{u}} = \arg \max_{\boldsymbol{u} \in \{0,1\}^k} \mathbb{P}[\boldsymbol{u}|\boldsymbol{y}]$, whose complexity grows exponentially in $k$ and is computationally infeasible. Given this, we compare our CRISP decoder with the SCL (Tal & Vardy, 2015), which has near-MAP performance for a large $L$, along with the classical SC. Among learning-based decoders, we choose the state-of-the-art Neural-Successive-Cancellation (NSC) as our baseline (Doan et al., 2018). While the original NSC uses a sub-optimal training procedure with SC probabilities as the target, we consider an improved version with end-to-end training (Fig. 2) for a fair comparison. We also include the performance of CRISP trained directly without the curriculum. We also compare with the curriculum training procedure of Lee et al. (2020) via the *C2N* scheme (Sec. 5, Fig. 11). All these baselines have the same number of parameters as CRISP.

Fig. 1a highlights that the CRISP decoder outperforms the existing baselines and attains near-MAP performance over a wide range of SNRs for the Polar$(64, 22)$ code. Fig. 1b illustrates the mechanism behind these gains at 0dB: the curriculum-guided CRISP slowly improves upon the overall BER (over the 22 bits) during the training and eventually achieves much better performance than SC and other baselines. In contrast, the decoder trained from scratch makes a big initial gain but gets stuck at local minima and only achieves a marginal improvement over SC. We observe a similar trend for Polar$(32, 16)$ code in Fig. 5a, where CRISP achieves near-MAP performance. We posit that aided by a good curriculum, CRISP avoids getting stuck at bad local minima and converges to better minima in the optimization landscape. App. D highlights similar reliability gains for non-AWGN channels, other blocklengths, and rates. App. C illustrates the ablation analysis

## 4.2 RELIABILITY-COMPLEXITY COMPARISON

Table 1: Throughput and reliability comparison of various decoders on Polar$(n, k)$.

| Decoder | Throughput (in Mbps) | | | | Gap to SCL, L=32 (in dB) | |
|---|---|---|---|---|---|---|
| | $(32, 16)$ | | $(64, 22)$ | | $(32, 16)$ | $(64, 22)$ |
| | GPU | CPU | GPU | CPU | | |
| SC | 0.17 | 27 | 0.08 | 15 | 0.80 | 0.40 |
| FastSC | N/A | **47** | N/A | **40** | 0.80 | 0.40 |
| SCL, L=4 | 0.01 | 8.5 | 0.02 | 6.27 | 0.05 | 0.10 |
| FastSCL, L=4 | N/A | **30** | N/A | **24** | 0.05 | 0.10 |
| SCL, L=32 (MAP) | 5e-3 | 0.81 | 2e-3 | 0.60 | **0.00** | **0.00** |
| FastSCL, L=32 | N/A | 7.7 | N/A | 5.5 | 0.00 | 0.00 |
| NSC | N/A | N/A | 32.6 | 0.02 | N/A | 0.35 |
| CRISP_GRU (Ours) | **80** | 0.04 | **38.7** | 0.02 | **0.15** | **0.20** |
| CRISP_CNN (Ours) | **250** | 0.02 | **133** | 0.13 | 0.15 | 0.20 |
| CRISP_GRU - No curriculum | 80 | 0.04 | 38.7 | 0.02 | 0.60 | 0.35 |

In the previous section, we demonstrated that CRISP achieves better reliability than the baselines. Here we analyze these gains through the lens of *decoding complexity*. To quantitatively compare the complexities of these decoders, we evaluate their throughput on a single GTX 1080 Ti GPU as well as a CPU (Intel i7-6850K, 12 threads). For the GPU version, we use our implementation of SC/SCL owing to the lack of publicly available implementations. On the other hand, for the CPU column we use an optimized multithreaded implementation of SC/FastSC, SCL/FastSCL (Léonardon et al. (2019)) in C++ by Cassagne et al. (2019). As Table 1 highlights, CRISP exploits the GPUs' inherent optimization towards NNs to achieve excellent throughput, whilst achieving near-SCL BER performance. We also design CRISP_CNN, a 1D-CNN decoder trained similar to CRISP_GRU (App. C.2), that attains better throughput than CRISP_GRU, while maintaining gains in BER. We posit that further improvement in throughput can be realized using techniques like pruning and knowledge distillation. This is beyond the scope of this paper and is an important and separate direction of future research. We refer to App. F for further discussion. Note that we use BER$= 10^{-3}$ to compute the gap to SCL (Figs. 1a, 5a)

## 4.3 INTERPRETATION

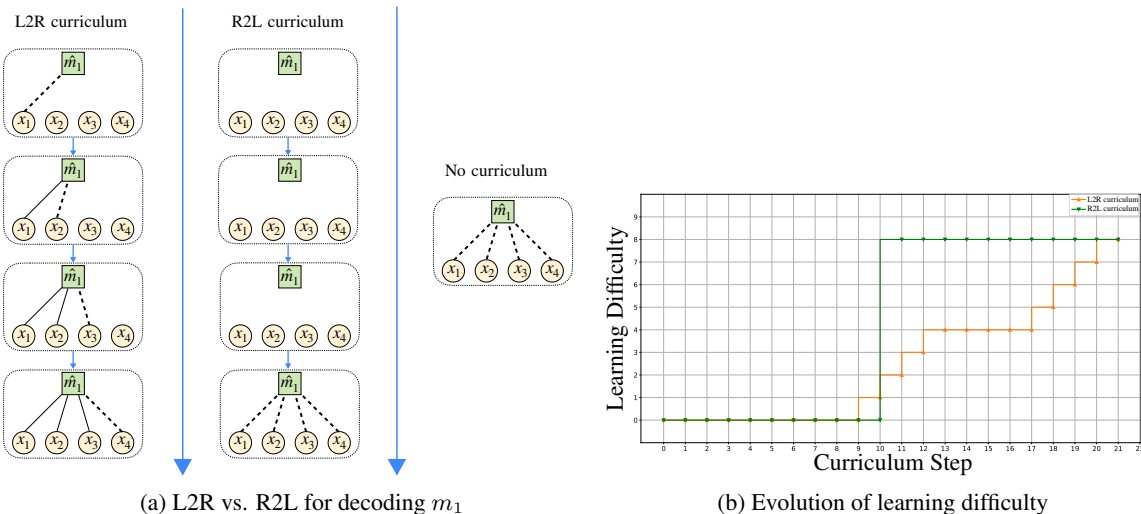

(a) L2R vs. R2L for decoding $m_1$        (b) Evolution of learning difficulty

Figure 6: L2R vs. R2L: (a) Bit estimates evolve more smoothly under L2R than R2L for Polar$(4, 4)$, (b) Learning difficulty increases more gracefully for L2R than R2L for Polar$(64, 22)$.

This section describes why L2R is a better curriculum than others. To this end, we first claim that learning to decode uncorrupted codewords ($\boldsymbol{y} = \boldsymbol{x}$) is critical to learning a reliable decoder. This claim follows from the following key observation: while training our model (sequential or block) at a specific SNR, we observe that whenever our model reaches SC or better performance, its BER on uncorrupted codewords, aka the noiseless BER, drops to zero very early in the training (App. B, Fig. 7a). On the other hand, when the model gets stuck at bad minima even after a lot of training, its noiseless BER is high (Fig. 7b). Hence, without loss of generality, we focus on the setting $\boldsymbol{y} = \boldsymbol{x}$. Under this noiseless scenario, we analyze how the optimal bit decoding rules evolve for different curricula. In particular, we focus on the least reliable bits as they contribute the largest to noiseless BER (Fig. 8a and Fig. 8b).

For the Polar$(4, 4)$ code, Fig. 6a illustrates how the optimal rule evolves for its least reliable bit $m_1$. In this case, the MAP decoding rule for $m_1$ is: $\hat{m}_1 = x_1 x_2 x_3 x_4$. Under the L2R curriculum, we arrive at this expression via $x_1 \rightarrow x_1 x_2 \rightarrow x_1 x_2 x_3 \rightarrow x_1 x_2 x_3 x_4$, whereas R2L follows $1 \rightarrow 1 \rightarrow 1 \rightarrow x_1 x_2 x_3 x_4$. This highlights that L2R reaches the optimal rule more gracefully by learning to include one coordinate $x_i$ at a time while this change for R2L (and no-curriculum) is abrupt, making it harder to learn. Fig. 9 illustrates a similar evolution for the remaining bits $(m_2, m_3, m_4)$.

More concretely, we define the notion of *learning difficulty* for a bit: the number of bits multiplied in its optimal decoding rule. This metric roughly captures the number of operations a model has to learn at any curriculum step. Fig. 6b illustrates how it evolves over the L2R and R2L curricula for the least reliable bit in Polar$(64, 22)$. If we take the maximum learning difficulty over all bits, we obtain a similar plot (Fig. 10). Note that in both the plots, the jumps in learning difficulty are larger for R2L, thus indicating a harder transfer than L2R, where it increases smoothly (at most one bit per step).

### 4.4 PAC CODES

A recent breakthrough work Arıkan (2019) introduces a new class of codes called Polarization-Adjusted-Convolutional (PAC) codes that match the fundamental lower bound on the performance of any code under the MAP decoding at finite-lengths (Moradi et al., 2020). The motivating idea behind PAC codes is to overcome two key limitations of polar codes at finite blocklengths: the poor minimum distance properties of the code and the sub-optimality of SC compared to the MAP (Mondelli et al., 2014). This is addressed by adding a *convolutional outer code*, with an appropriate indexing $I_k$, before polar encoding to improve the distance properties of the resulting code. More formally, the message block $\boldsymbol{u} \in \{0, 1\}^k$ is embedded into the source vector $\boldsymbol{m} \in \{0, 1\}^n$ according to the Reed-Muller (RM) indices $I_k^{\text{(RM)}}$: compute the Hamming weights of integers $0, 1, \ldots, n-1$ and choose the top $k$. Now we encode the message $\boldsymbol{m}$ via a rate-1 convolutional code, i.e. $\boldsymbol{v} = \boldsymbol{c} * \boldsymbol{m} \in \{0, 1\}^n \Leftrightarrow v_i = \sum_j c_j m_{i-j}$, for some 1D convolutional kernel $\boldsymbol{c} \in \{0, 1\}^\ell$. Finally we obtain the PAC codeword $\boldsymbol{x}$ by polar encoding $\boldsymbol{v}$: $\boldsymbol{x} = \text{PlotkinTree}(\boldsymbol{v})$.

PAC codes can be decoded using the classical Fano algorithm (Fano, 1963), a sequential decoding algorithm that uses backtracking. Coupled with the Fano decoder, PAC codes achieve impressive results outperforming polar codes (with SCL decoder) and matching the finite-length capacity bound (Polyanskiy et al., 2010). However, the Fano decoder has significant drawbacks like variable running time, large time complexity at low-SNRs (Rowshan et al., 2020b), and sensitivity to the choice of hyperparameters (Moradi, 2020). To overcome these issues, several non-learning techniques, such as stack/list decoding, adaptive path metrics, etc., have been proposed in the literature (Yao et al., 2021; Zhu et al., 2020; Rowshan & Viterbo, 2021b;a; Sun et al., 2021). In contrast, we design a curriculum-learning-based CRISP decoder for PAC codes trained directly from the data. We use the same L2R curriculum to decode PAC codes.

Fig. 5b highlights that the CRISP decoder achieves near-MAP performance for the PAC$(32, 16)$ code. While Fano decoding achieves similar reliability, it is inherently non-parallelizable. In contrast, CRISP allows for batching, and achieves a higher throughput, as highlighted in Table 2. Here we measure the throughput of Fano (Rowshan et al., 2020a) at SNR $= 1$ dB. We note that the existing implementation of Fano is not supported on GPUs. These preliminary results suggest that curriculum-based training holds a great promise for designing efficient PAC decoders, especially for longer blocklengths, which is an interesting topic of future research (App. D.2).

Table 2: Throughput and reliability comparison of various decoders on PAC$(32, 16)$.

| Decoder | Throughput (in Mbps) | | Gap to SCL, L=128 (in dB) |
|---|---|---|---|
| | GPU | CPU | |
| SC | N/A | **27** | 1.0 |
| SCL, L=128 | N/A | 0.02 | **0.0** |
| Fano | N/A | 4e-3 | 0.1 |
| CRISP_GRU (Ours) | **80** | 0.03 | 0.4 |
| CRISP_CNN (Ours) | **250** | 0.15 | 0.4 |
| CRISP_GRU - No curriculum | 80 | 0.03 | 0.8 |

## 5 INFORMATION THEORY GUIDED CURRICULA

In Sec. 4, we demonstrated the superiority of L2R curriculum over other schemes. Here we introduce an alternate curriculum, *Noisy-to-Clean (N2C)*, that slightly bests the L2R, inspired by the polarization property of polar codes. The key idea behind N2C curriculum is to capitalize on the polar index set $I_k$. Recall that the set $I_k$ is obtained by ranking the $n$ polar bit channels (under SC decoding) in the increasing order of their reliabilities (from noisy to clean) and choosing the top $k$ indices. Formally, given $I_k = \{i_{r1}, i_{r2}, \ldots, i_{rk}\} \subseteq [n]$ in the increasing order of reliabilities, our *N2C* curriculum is given by: Train $\theta$ on $\hat{m}_{i_{r1}} \to$ Train $\theta$ on $(\hat{m}_{i_{r1}}, \hat{m}_{i_{r2}}) \to \ldots \to$ Train $\theta$ on $(\hat{m}_{i_{r1}}, \ldots, \hat{m}_{i_{rk}})$. For both the sequential and block decoders, we observe that N2C is the best curriculum and we have N2C $\approx$ L2R > C2N $\approx$ R2L (Fig. 11). This ordering is consistent with our interpretation in Sec. 4.3 of how the learning difficulty evolves over a curriculum (Fig. 12). For both N2C and L2R, the learning difficulty evolves smoothly but is abrupt for C2N and R2L, thus making transfer harder in these curricula. Note that the *C2N* curriculum refers to progressively training on subcodes of Polar$(n, k)$: Polar$(n, 1) \to \ldots \to$ Polar$(n, k)$ (Lee et al., 2020).

## 6 RELATED WORK

To address the sub-optimality of SC at finite lengths, a popular technique is to use list decoding (Tal & Vardy, 2015; Balatsoukas-Stimming et al., 2015), aided by cyclic redundancy checks (CRC) (Li et al., 2012; Niu & Chen, 2012a; Miloslavskaya & Trifonov, 2014). Several alternate decoding methods have also been proposed such as stack decoding (Niu & Chen, 2012b; Trifonov, 2018), belief propagation decoding (Yuan & Parhi, 2014; Elkelesh et al., 2018). Deep learning for communication (Qin et al., 2019; Kim et al., 2020) has been an active field in the recent years and has seen success in many problems including the design of neural decoders for existing linear codes (Nachmani et al., 2016; O'shea & Hoydis, 2017; Lugosch & Gross, 2017; Vasić et al., 2018; Liang et al., 2018; Bennatan et al., 2018; Jiang et al., 2019a; Nachmani & Wolf, 2019; Buchberger et al., 2020; He et al., 2020), and jointly learning channel encoder-decoder pairs. (O'Shea et al., 2016; Kim et al., 2018a; Jiang et al., 2019b; Makkuva et al., 2021; Jamali et al., 2021; Chahine et al., 2021a;b).

Earlier works on designing neural polar decoders (Gross et al., 2020) used off-the-shelf neural architectures. These were only able to decode codes of small blocklength ($\leq 16$) (Lyu et al., 2018; Cao et al., 2020). Later works augmented belief propagation decoding (Xu et al., 2018; Doan et al., 2019), with neural components and improved performance. In Cammerer et al. (2017a) and Doan et al. (2018), the authors replace sub-components of the existing SC decoder with NNs to scale decoding to longer lengths. However, these methods fail to give reasonable reliability gains compared to SC. In contrast, we use curriculum learning to train neural decoders, and show non-trivial gains over SC performance. Our approach is closest to that of Lee et al. (2020), who consider curriculum training of polar decoder via the *C2N* scheme, upon which we strictly improve.

## 7 CONCLUSION

We introduce a novel curriculum based neural decoder, CRISP, that attains near-optimal reliability on the Polar code family in the short blocklength regime. Extending our results to medium blocklengths (100-1000) and codes outside the Polar family are interesting future directions.

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

# A  SUCCESSIVE CANCELLATION DECODER

Here we detail the successive-cancellation (SC) algorithm for decoding polar codes. As a motivating example, let's consider the $\text{Polar}(2, 2)$ code. Let the two information bits be denoted by $u$ and $v$, where $u, v \in \{0, 1\}$. The codeword $\boldsymbol{x} \in \{0, 1\}^2$ is given by $\boldsymbol{x} = (x_1, x_2) = (u \oplus v, v)$. Let $\boldsymbol{y} \in \mathbb{R}^2$ be the corresponding noisy codeword received by the decoder. First we convert the received $\boldsymbol{y}$ into a vector of log-likelihood-ratios (LLRs), $\boldsymbol{L_y} \in \mathbb{R}^2$, which contains the soft-information about coded bits $x_1$ and $x_2$, i.e.

$$\boldsymbol{L_y} = (\boldsymbol{L_y^{(1)}}, \boldsymbol{L_y^{(2)}}) \triangleq \left( \log \frac{\mathbb{P}[y_1 | x_1 = 0]}{\mathbb{P}[y_1 | x_1 = 1]}, \log \frac{\mathbb{P}[y_2 | x_2 = 0]}{\mathbb{P}[y_2 | x_2 = 1]} \right) \in \mathbb{R}^2.$$

Once we have the soft-information about the codeword $\boldsymbol{x}$, the goal is to now obtain the same for the message bits $u$ and $v$. To compute the LLRs for these information bits, SC uses the following principle: first, compute the soft-information for the left bit $u$ to estimate $\hat{u}$. Use the decoded $\hat{u}$ to compute the soft-information for the right bit $v$ and decode it. More concretely, we compute the LLR for the bit $u$ as:

$$L_u = \text{LSE}(\boldsymbol{L_y^{(1)}}, \boldsymbol{L_y^{(2)}}) = \log \frac{1 + e^{\boldsymbol{L_y^{(1)}} + \boldsymbol{L_y^{(2)}}}}{e^{\boldsymbol{L_y^{(1)}}} + e^{\boldsymbol{L_y^{(2)}}}} \in \mathbb{R}, \tag{6}$$

where $\text{LSE}(a, b) \triangleq \log(1 + e^{a+b})/(e^a + e^b)$ for $a, b \in \mathbb{R}$. The expression in Eq. 6 follows from the fact that $u = (u \oplus v) \oplus v = x_1 \oplus x_2$ and hence the soft-information $L_u$ can be accordingly derived from that of $x_1$ and $x_2$, i.e. $\boldsymbol{L_y}$. Now we estimate the bit as $\hat{u} = \mathbb{1}\{L_u < 0\}$. Assuming that we know the bit $u = \hat{u}$, we observe that the codeword $\boldsymbol{x} = (\hat{u} \oplus v, v)$ can be viewed as a two-repitition of $v$. Hence its LLR $L_v$ is given by

$$L_v = \boldsymbol{L_y^{(1)}} \cdot (-1)^{\hat{u}} + \boldsymbol{L_y^{(2)}} \in \mathbb{R}. \tag{7}$$

Finally we decode the bit as $\hat{v} = \mathbb{1}\{L_v < 0\}$. To summarize, given the LLR vector $\boldsymbol{L_y}$ we first compute the LLR for the bit $u$, $L_u$, using Eq. 6 and decode it. Utilizing the decoded version $\hat{u}$, we compute the LLR $L_v$ according to Eq. 7 and decode it.

For a more generic $\text{Polar}(n, k)$, the underlying principle is the same: to decode a polar codeword $\boldsymbol{x} = (\boldsymbol{u} \oplus \boldsymbol{v}, \boldsymbol{v})$, first decode the left child $\boldsymbol{u}$ and utilize this to decode the right child $\boldsymbol{v}$. This principle is recursively applied at each node of the Plotkin tree until we reach the leaves of the tree where the decoding is trivial. In view of this principle, the SC algorithm for $\text{Polar}(2, 4)$, illustrated in Fig. 3b, can be mathematically expressed as (in the sequence of steps):

$$\boldsymbol{y} \in \mathbb{R}^4 \longrightarrow \boldsymbol{L_y} = (\boldsymbol{L_y^{(1)}}, \boldsymbol{L_y^{(2)}}, \boldsymbol{L_y^{(3)}}, \boldsymbol{L_y^{(4)}}) \in \mathbb{R}^4,$$
$$\boldsymbol{L_u} = (\text{LSE}(\boldsymbol{L_y^{(1)}}, \boldsymbol{L_y^{(3)}}), \text{LSE}(\boldsymbol{L_y^{(2)}}, \boldsymbol{L_y^{(4)}})) \in \mathbb{R}^2,$$
$$\text{frozen bit} \longrightarrow \hat{m}_1 = 0,$$
$$L_2 = \text{LSE}(\boldsymbol{L_y^{(1)}}, \boldsymbol{L_y^{(3)}}) + \text{LSE}(\boldsymbol{L_y^{(2)}}, \boldsymbol{L_y^{(4)}}) \in \mathbb{R},$$
$$\hat{m}_2 = \mathbb{1}\{L_2 < 0\} \in \{0, 1\},$$
$$\hat{\boldsymbol{u}} = (\hat{m}_2, \hat{m}_2) \in \{0, 1\}^2,$$
$$\boldsymbol{L_v} = (\boldsymbol{L_y^{(1)}}, \boldsymbol{L_y^{(2)}}) \cdot (-1)^{\hat{\boldsymbol{u}}} + (\boldsymbol{L_y^{(3)}}, \boldsymbol{L_y^{(4)}}) \in \mathbb{R}^2,$$
$$\text{frozen bit} \longrightarrow \hat{m}_3 = 0,$$
$$L_4 = \boldsymbol{L_v^{(1)}} + \boldsymbol{L_v^{(2)}} \in \mathbb{R},$$
$$\hat{m}_4 = \mathbb{1}\{L_4 < 0\} \in \{0, 1\}.$$

In Fig. 3b, the above equations are succinctly represented by two set of arrows: the black solid arrows represent the flow of soft-information from the parent node to the children whereas the green dotted arrows represent the flow of the decoded bit information from the children to the parent. We note that we use a simpler min-sum approximation for the function LSE that is often used in practice, i.e.

$$\text{LSE}(a, b) \approx \min(|a|, |b|)\text{sign}(a)\text{sign}(b), \quad a, b \in \mathbb{R}.$$

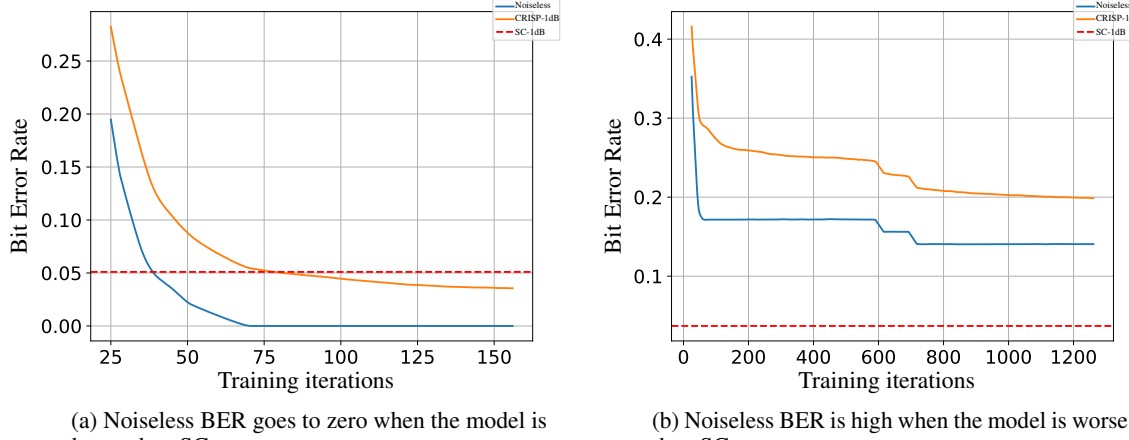

Figure 7: Evolution of training BER at 1dB and noiseless BER for CRISP.

(a) Noiseless BER goes to zero when the model is better than SC

(b) Noiseless BER is high when the model is worse than SC

## B  INTERPRETATION

As discussed in Sec. 4.3, we observe that whenever our decoder reaches SC or better performance eventually when training at a specific SNR, its BER (over all the bits) on uncorrupted codewords, noiseless BER, drops to $0$ early on in the training. Fig. 7a illustrates this for $\text{Polar}(32, 16)$. Conversely, if the model gets stuck at a BER worse than that of SC, then we observe that its noiseless BER is also stuck at a non-zero value. This is highlighted in Fig. 7b for $\text{Polar}(64, 32)$. In particular, we notice that the least reliable bits contribute the most to the noiseless BER, while a majority of the cleaner bits have zero individual BER (Fig. 8a). Viewed from this context, we focus on the noiseless scenario, i.e. $\boldsymbol{y} = \boldsymbol{x}$.

As a motivating example, we first consider the $\text{Polar}(4, 4)$ code. Let $\boldsymbol{m} = (m_1, m_2, m_3, m_4) \in \{0, 1\}^4$ be the block of message bits and $\boldsymbol{x} \in \{0, 1\}^4$ be the codeword. Hence under the L2R curriculum, the subcodes evolve as

- $k = 1 : m_1 \mapsto (m_1, 0, 0, 0) \mapsto \boldsymbol{x} = (m_1, 0, 0, 0)$,
- $k = 2 : (m_1, m_2) \mapsto (m_1, m_2, 0, 0) \mapsto \boldsymbol{x} = (m_1 \oplus m_2, m_2, 0, 0)$,
- $k = 3 : (m_1, m_2, m_3) \mapsto (m_1, m_2, m_3, 0) \mapsto \boldsymbol{x} = (m_1 \oplus m_2 \oplus m_3, m_2, m_3, 0)$,
- $k = 4 : (m_1, m_2, m_3, m_4) \mapsto (m_1, m_2, m_3, m_4) \mapsto \boldsymbol{x} = (m_1 \oplus m_2 \oplus m_3 \oplus m_4, m_2 \oplus m_4, m_3 \oplus m_4, m_4)$.

Correspondingly, their optimal bit decoding rules under the MAP evolve as

- $k = 1 : \boldsymbol{y} = \boldsymbol{x} = (m_1, 0, 0, 0) \mapsto \hat{m}_1 = x_1$,
- $k = 2 : \boldsymbol{y} = \boldsymbol{x} = (m_1 \oplus m_2, m_2, 0, 0) \mapsto (\hat{m}_1, \hat{m}_2) = (x_1 \oplus x_2, x_2)$,
- $k = 3 : \boldsymbol{y} = \boldsymbol{x} = (m_1 \oplus m_2 \oplus m_3, m_2, m_3, 0) \mapsto (\hat{m}_1, \hat{m}_2, \hat{m}_3) = (x_1 \oplus x_2 \oplus x_3, x_2, x_3)$,
- $k = 4 : \boldsymbol{y} = \boldsymbol{x} = (m_1 \oplus m_2 \oplus m_3 \oplus m_4, m_2 \oplus m_4, m_3 \oplus m_4, m_4) \mapsto (\hat{m}_1, \hat{m}_2, \hat{m}_3, \hat{m}_4) = (x_1 \oplus x_2 \oplus x_3 \oplus x_4, x_2 \oplus x_4, x_3 \oplus x_4, x_4)$.

Similarly, we can compute the subcodes and their corresponding decision rules under the R2L curriculum. Fig. 9 illustrates this evolution for both L2R and R2L. For the least reliable bit $m_1$, we observe that the L2R curriculum reaches the optimal rule more gracefully by including one coordinate $x_i$ at a time while this change for R2L (and no-curriculum) is abrupt, making it harder to learn. We observe the same trend for other bits $m_2$, $m_3$ and $m_4$. Note that for $\text{Polar}(4, 4)$, the reliability order is $m_1 < m_2 = m_3 < m_4$ and hence the L2R curriculum is same as N2C and R2L is same as C2N.

For a general $\text{Polar}(n, k)$, we can likewise compute the optimal MAP rules using the fact that the mapping $\text{PlotkinTree} : \{0, 1\}^n \to \{0, 1\}^n$ is its own inverse, i.e. $\boldsymbol{x} = \text{PlotkinTree}(\boldsymbol{m}) \implies \boldsymbol{m} = \text{PlotkinTree}(\boldsymbol{x})$.

To concretely compare different curricula, we define the notion of *learning difficulty* for a bit: the number of codeword bits multiplied in its optimal decoding rule. This metric roughly captures the number of multiplication operations a model has to learn at any curriculum step. For example, for Polar$(4, 4)$, the learning difficulty for $m_1$ evolves as $1 \to 2 \to 3 \to 4$ for the L2R curriculum and as $0 \to 0 \to 0 \to 4$ for the R2L curriculum. Fig. 10 illustrates the evolution of learning difficulty (taking maximum over all bits) for Polar$(32, 16)$ and Polar$(64, 22)$ codes. We observe here that the jumps in the learning difficulty are larger for R2L, thus indicating a harder transfer than L2R, where it increases smoothly (at most one bit per step).

Fig. 12 highlights a similar phenomenon for Polar$(64, 22)$ for L2R, R2L, N2C and C2N curricula. We observe that the learning difficulties of the L2R and N2C curricula evolve smoothly while that of R2L and C2N are abrupt. Correspondingly, their final BER reliability performance follows the order N2C $\approx$ L2R < R2L $\approx$ C2N (Fig. 11).

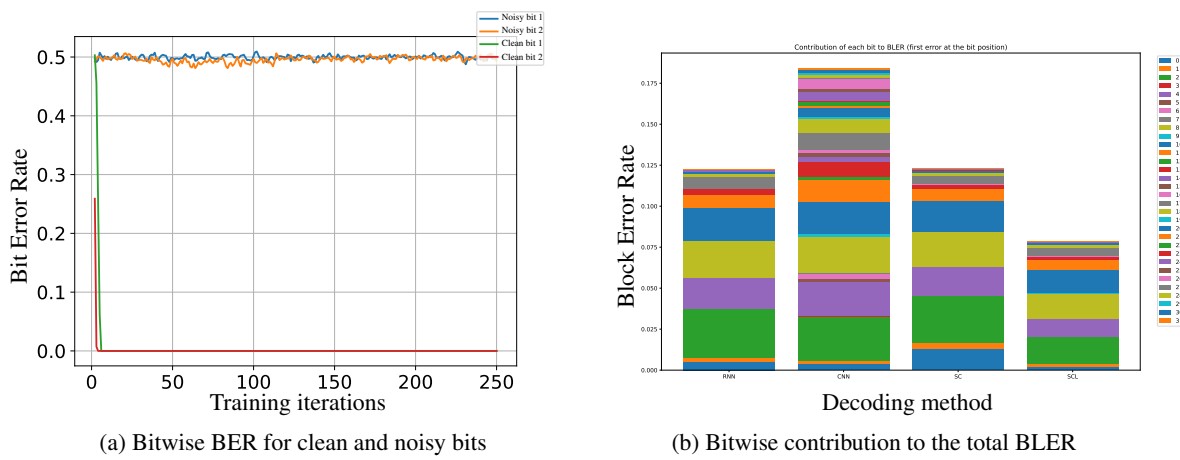

(a) Bitwise BER for clean and noisy bits

(b) Bitwise contribution to the total BLER

Figure 8: Error analysis for Polar$(64, 32)$ : (a) Noiseless BER for the two least reliable bits gets stuck at $0.5$ whereas it converges to $0$ for the two most reliable bits, (b) Contribution of each bit (conditioned on no previous errors) to the BLER.

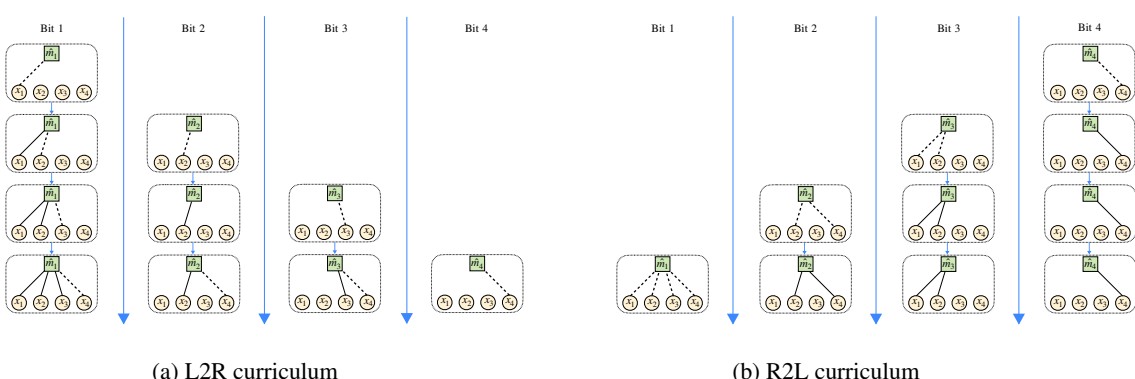

(a) L2R curriculum

(b) R2L curriculum

Figure 9: Evolution of the MAP decoding rules for L2R and R2L for Polar$(4, 4)$. Dotted lines indicate new coded bits being introduced into the decoding rule at each curriculum step.

## C  ABLATION STUDIES

Recall that our CRISP decoder consists of the sequential RNN (512-dim hidden state) trained with the L2R curriculum. To understand the contribution of each of these components to its gains over SC, we did the following ablation experiments for Polar$(64, 22)$ code.

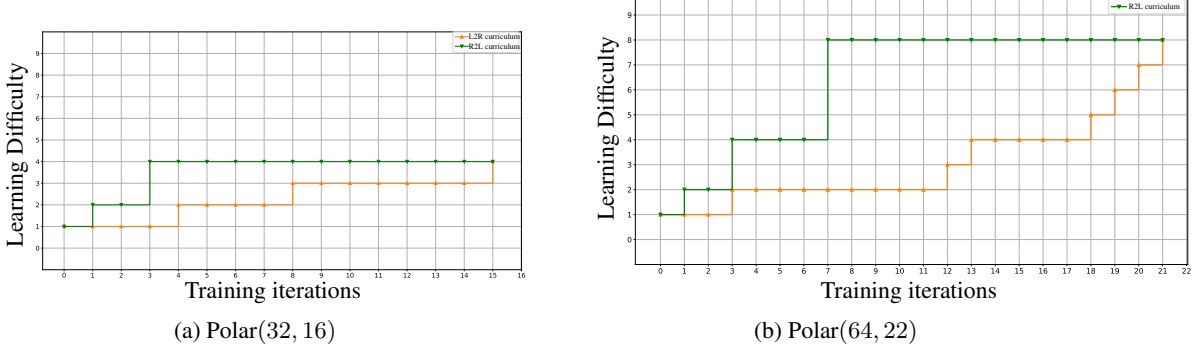

(a) Polar(32, 16)  (b) Polar(64, 22)

Figure 10: Evolution of the learning difficulty for L2R and R2L.

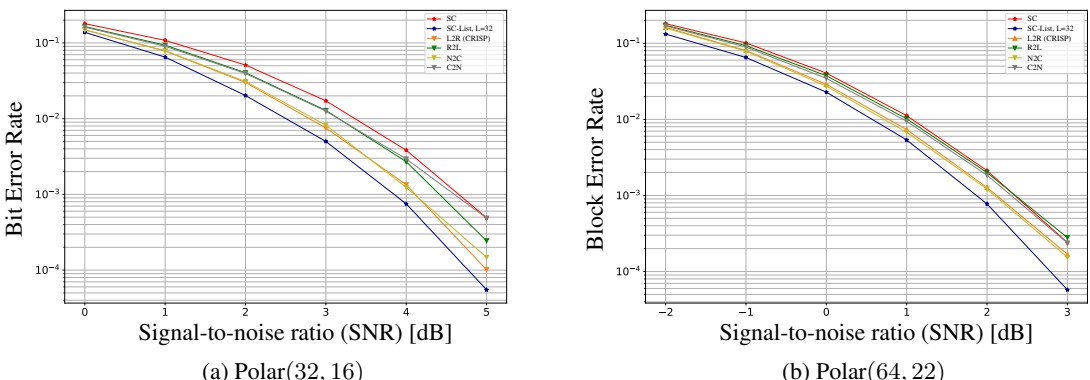

(a) Polar(32, 16)  (b) Polar(64, 22)

Figure 11: Information-theory guided curricula N2C and C2N are marginally better than the L2R and R2L schemes respectively.

### C.1  EFFECT OF MODEL SIZE

We fix the decoder to be GRU and consider different model sizes via the hidden state size $h \in \{256, 512\}$, and different curricula among {L2R, R2L, Without curriculum (w/o C)}. Fig. 13a demonstrates that the accuracy gains of the L2R curriculum are more pronounced for *smaller* models ($h = 256$). On the other hand, we observe minimal reliabilty gains for L2R with large models ($h = 512$). We also tried other sequential architectures such as LSTMs (Hochreiter & Schmidhuber, 1997) and Transformers (Radford et al., 2019), but found GRUs to be the best (App. D).

### C.2  SEQUENTIAL VS. BLOCK DECODING

We note that the sequential RNN architecture for CRISP is inspired in part by the sequential SC algorithm. Notwithstanding, we also designed block decoders that estimate all the information bits $m_i$ in one shot given $\boldsymbol{y}$. We choose 1D Convolutional Neural Networks (CNNs) to parametrize this block decoder. Similar to sequential decoders, curriculum learning; in particular, the L2R scheme works the best for block decoding in achieving near-MAP reliability.

Fig. 14b compares RNNs and CNNs in terms of BLER for Polar(64, 22) with L2R and R2L curricula. We observe that RNN-based decoders (CRISP) are more reliable in terms of BLER than CNNs; in contrast, RNNs and CNNs achieve similar BER performance (Fig. 14a). Further, we observe that the error patterns corresponding to bitwise contribution to the total BLER for the RNN model resemble that of SC-List, as opposed to CNN models (Fig. 8b). We show the evolution of validation BER for CNN training in Fig. 15. We see that the C2N curriculum performs worse than the N2C curriculum.

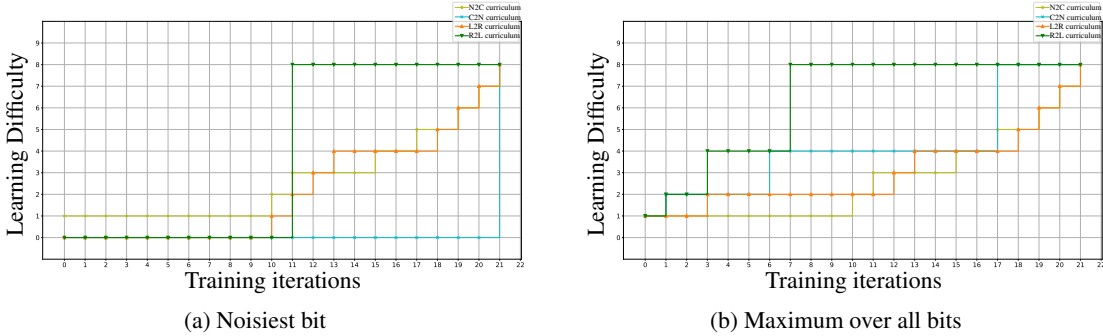

(a) Noisiest bit

(b) Maximum over all bits

Figure 12: Evolution of learning bit difficulty for different curricula for Polar$(64, 22)$.

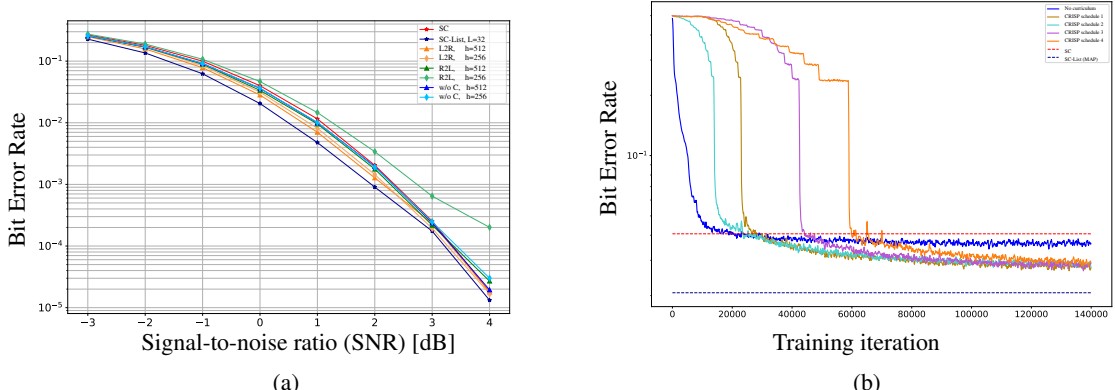

(a)

(b)

Figure 13: Ablation plots: (a) Choosing the right curriculum is critical when model size is smaller, (b) The number of iterations to train CRISP on each subcode using the L2R/N2C curriculum are not critical to the final performance achieved.

## D  ADDITIONAL RESULTS

We present our additional results on the Polar code family with various decoding architectures such as CNNs and transformers, with BLER reliability, and for longer blocklengths ($n = 128$). Recall that the CRISP decoder uses the GRU-based RNN (Fig. 4a) trained with the L2R curriculum.

### D.1  ADDITIONAL RESULTS FOR POLAR CODES

#### D.1.1  ROBUSTNESS TO NON-AWGN NOISE

In this section we evaluate CRISP trained on AWGN on non-AWGN settings. We test CRISP on a Rayleigh fading channel, and T-distributed noise. As shown in Fig. 16, CRISP retains its gains when tested on a Rayleigh fading channel. Further, as demonstrated in Fig. 17, CRISP is very robust to T-distributed noise and marginally outperforms SCL at higher SNRs.

#### D.1.2  CRISP FOR CRC-POLAR CODES

In practice, polar codes with successive cancellation list decoding is used in conjunction with a cyclic redundancy check (CRC) outer code. The message $u \in \{0,1\}^{k_m}$ is encoded by a systematic cyclic code of rate $\frac{k_m}{k}$ to obtain a vector $m \in \{0,1\}^k$. We obtain the codewords via the normal polar encoding procedure on $m$. CRISP can be used to decode such CRC-Polar codes by considering $m$ as the input to the polar code block. As shown in Fig. 18, CRISP achieves near-MAP reliability when CRCs of length 3 and 8 are used for a Polar$(64, 22)$ code.

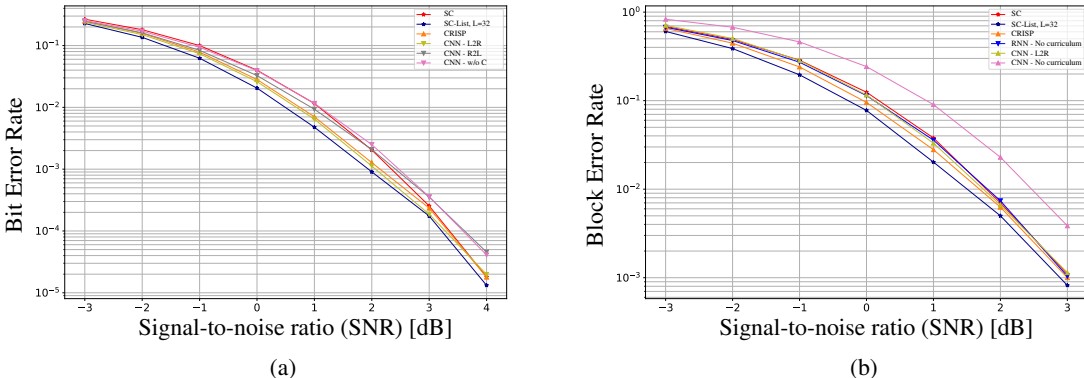

(a)             (b)

Figure 14: a) CNN decoder achieves near-MAP BER performance with L2R curriculum.
b) CRISP achieves near-MAP BLER for Polar$(64, 22)$. CNN is slightly worse.

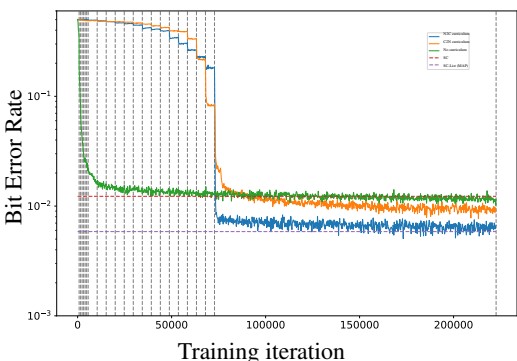

Figure 15: L2R curriculum helps CNN to achieve near-optimal reliability

### D.1.3 Scaling to larger codes

Curriculum training can be used to train even larger codes and obtain gains over naive training methods. However, we observed that our models were only able to achieve a reliability marginally better than SC. As shown in Fig. 21, CRISP performs similar to SC decoding on the Polar$(128, 22)$ code. We believe that it is possible to close the gap with MAP with more training tricks.

### D.1.4 Results with transformers

We also experimented with transformer-based architecures Vaswani et al. (2017) for our decoder. In particular, we tried an autoregressive transformer-decoder network (similar to GPT (Brown et al., 2020) that does sequential decoding) and the transformer-encoder network (similar to BERT (Devlin et al., 2018) that does block decoding). Preliminary results indicate that these transformer-based models are less reliable compared to RNNs and CNNs (Fig. 19). In addition, these models take a greater number of iterations (E) to train on each of the subcodes than RNNs and CNNs during curriculum training. Transformer training is sensitive to architectural and hyperparameter choices and is computationally expensive. We believe that with the right training tricks, transformer-based models can be used to decode larger codes. This is ongoing work.

### D.2 Additional results for PAC codes

CRISP maintains its good performance even in block error rate, as we show in Fig. 20b. Fig. 20a compares RNNs and CNNs in terms of BER for PAC$(32, 16)$ code with L2R and R2L curricula.

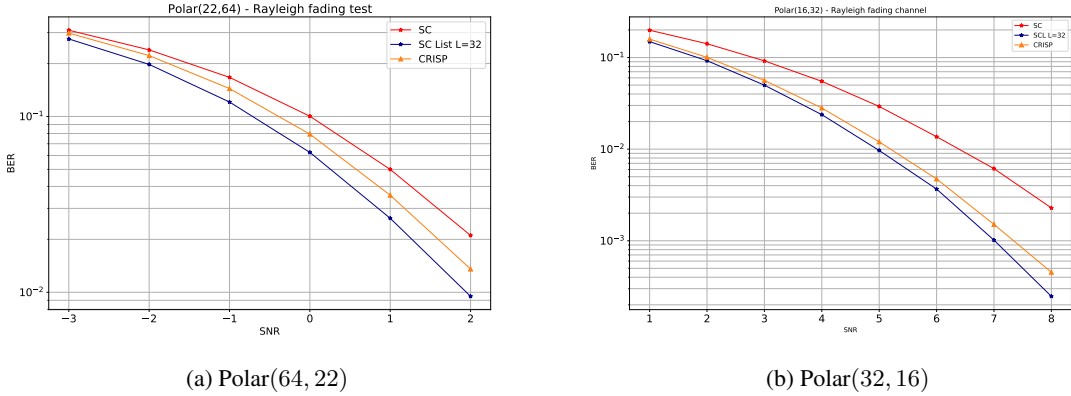

(a) Polar(64, 22)            (b) Polar(32, 16)

Figure 16: CRISP achieves good reliability on Rayleigh fading channels

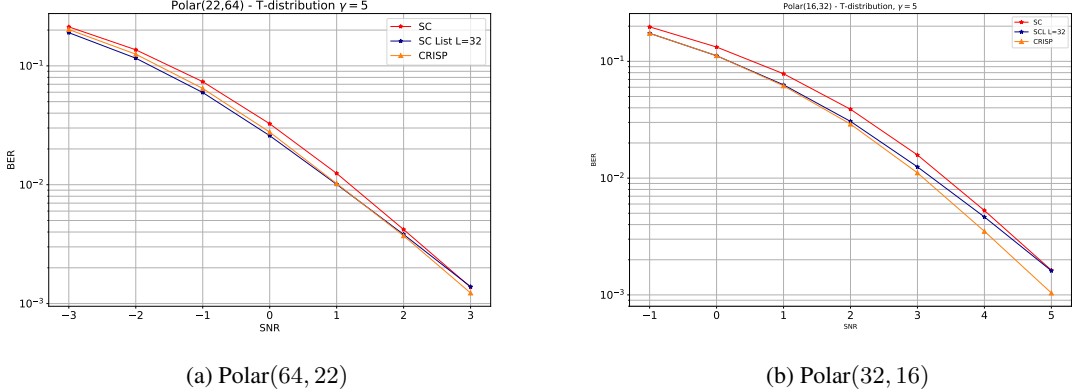

(a) Polar(64, 22)            (b) Polar(32, 16)

Figure 17: CRISP matches SCL reliability on T-distributed channels

We observe that while both RNNs and CNNs outperform SC, RNNs achieve slightly better BER reliability than CNNs. On the other hand, Fig. 20b highlights that CNNs achieves an SC-like BLER.

## E   EXPERIMENTAL DETAILS

We provide our code at the following link.

**Data generation.** Note that for any Polar$(n, k)$ or PAC$(n, k)$ code, the input message $m$ is chosen uniformly at random from $\{0, 1\}^k$. We simulate this by drawing $k$ i.i.d. Bernoulli random variables with probability $1/2$. We follow a similar procedure to generate a batch of message blocks (in $\{0, 1\}^{B \times k}$) with batch size $B$, both during training and inference. For the AWGN channel, the batch noise (in $\mathbb{R}^{B \times n}$) is accordingly generated by drawing i.i.d. Gaussian samples from $\mathcal{N}(0, \sigma^2)$.

**Hyper-parameters.** For training our models (both sequential and block decoders), we use AdamW optimizer (Loshchilov & Hutter, 2017) with a learning rate of $10^{-3}$. At each curriculum step, corresponding to training a subcode, we choose the SNR corresponding to which the optimal decoder for that subcode has BER in the range of $10^{-2} \sim 10^{-1}$ (Kim et al., 2018b). This ensures that a significant portion of training examples lie close to the decision boundary. It is well known that using a large batch size is essential to train a reliable decoder (Jiang et al., 2019a); we use a batch size of 4096 or 8192.

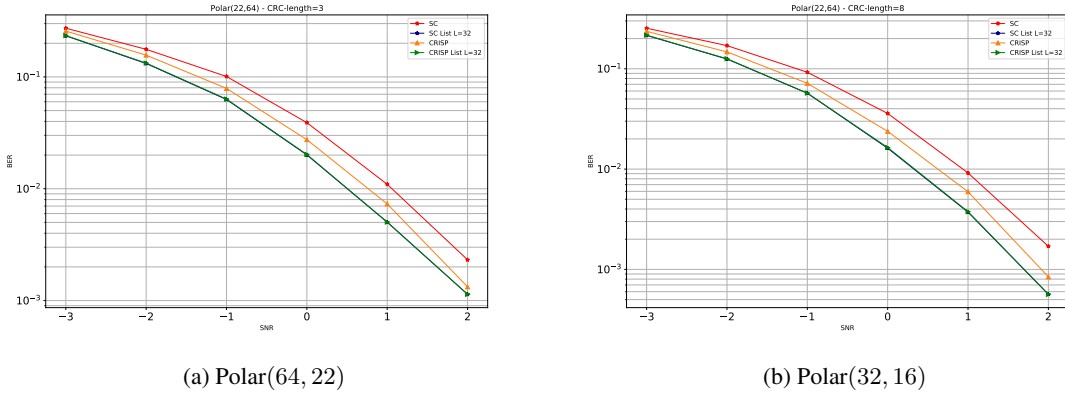

(a) Polar(64, 22)  (b) Polar(32, 16)

Figure 18: CRISP performs well on CRC-Polar code

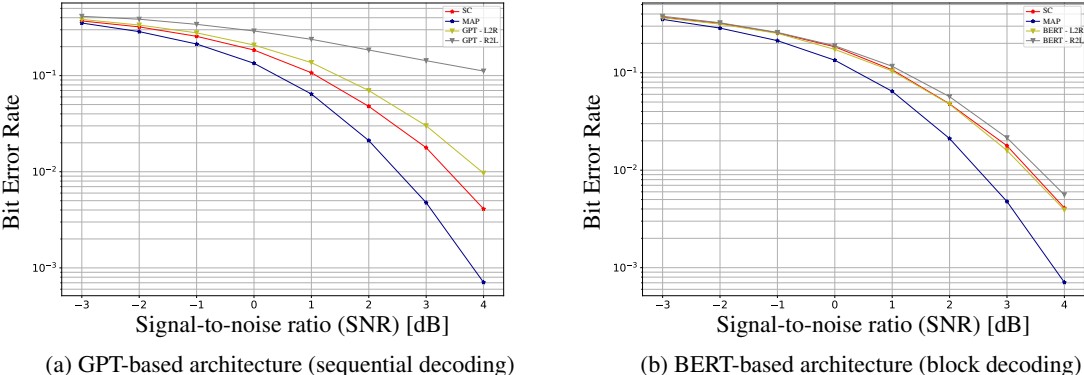

(a) GPT-based architecture (sequential decoding)  (b) BERT-based architecture (block decoding)

Figure 19: Transformer performance on Polar(32, 16).

### E.1 SEQUENTIAL DECODERS

We present the architectures and training details for our sequential decoders. We consider two popular choices for our sequential models: RNNs and GPT. We also note that it is a standard practice to use *teacher forcing* to train sequential models (Lamb et al., 2016): during training, as opposed to feeding the model prediction $\hat{m}_i$ as an input for the next time step, the ground truth message bit $m_i$ is provided as an input to the model instead (Fig. 4a). *Student forcing* refers to using the same $\hat{m}_i$ as an input.

#### E.1.1 RNNS

**Architecture.** We use a 2-layer GRU with a hidden state size of 512. The output at each timestep is obtained through a fully connected layer (as shown in Fig. 4a). The network has 2.5M and 600K parameters for block lengths 64 and 32. As shown in Figure 22, 2-layer-LSTM and 3-layer-GRU models achieve similar performance. We choose a 2-layer GRU for our experiments since it allows for faster training and has fewer parameters.

**Training.** We use the teacher forcing mechanism to train our models. We found that teacher forcing gives a better final performance in terms of both BER and BLER, whereas student forcing only provides gains in the BER reliability (Fig. 23). We observed that student forced training achieved suboptimal performance for larger block lengths. Empirically we observed that the number of iterations spent on training each intermediate subcode of the curriculum is not critical to the performance of the final model (Fig. 13b). To train CRISP for Polar(64,22), we use the following curriculum schedule: Train each subcode for 2000 iterations, and finally train the full code until convergence

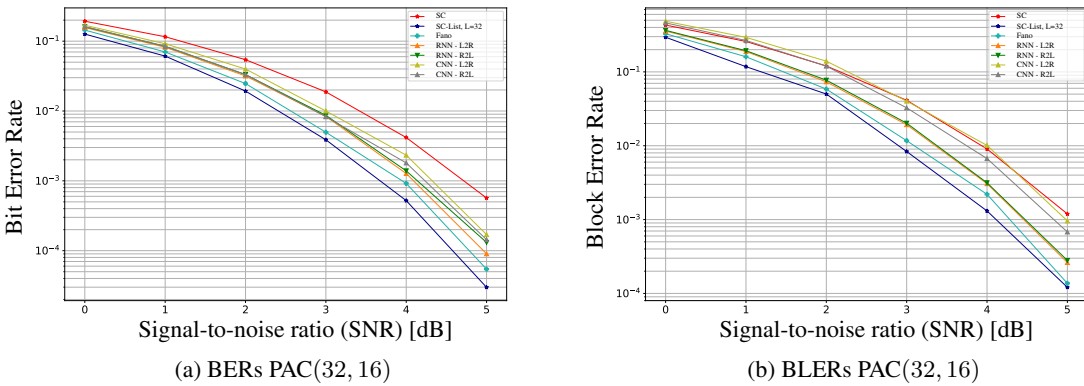

(a) BERs PAC(32, 16)  (b) BLERs PAC(32, 16)

Figure 20: With correct choice of curriculum, CNNs match the BER performance of CRISP on PAC(32,16). However, they are sub-optimal in BLER.

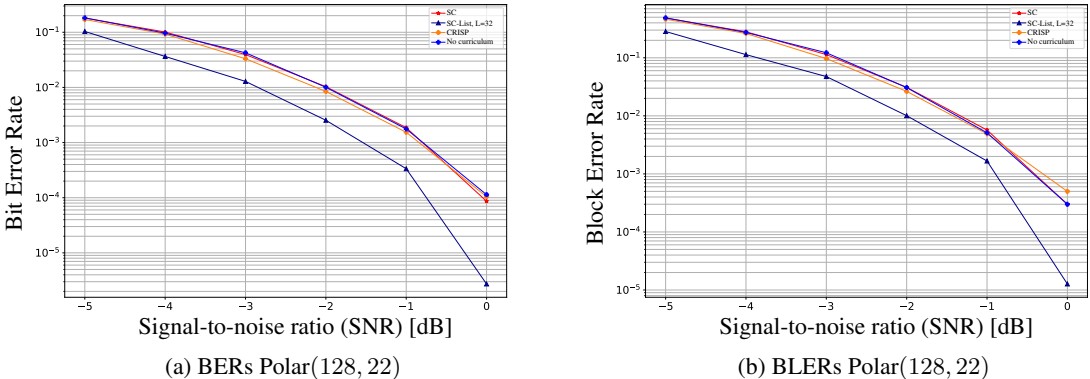

(a) BERs Polar(128, 22)  (b) BLERs Polar(128, 22)

Figure 21: CRISP matches SC reliabilty on Polar(128, 22).

with a decaying learning rate. This training schedule required 13-15 hours of training on a GTX 1080Ti GPU.

### E.1.2  GPT

**Architecture.** The model consists of 6 transformer blocks with masked self-attention and GELU activation. The multiheaded attention unit has 8 heads in each block, and an embedding/hidden size of 64 is used throughout the network. The output vectors of the final transformer block are passed through a linear layer to estimate each bit sequentially. The model has 350K parameters for blocklength 32.

**Training.** For training the GPT-based transformer, we use a teacher forcing mechanism. Here, we observed that the decoder takes a greater number of iterations $(40, 000)$ to train on each of the subcodes than RNNs and CNNs $(2, 000 - 10, 000)$ during curriculum training of Polar$(32, 16)$. For a fixed batch size, GPT also takes significantly longer to train (12 hours) compared to CNNs (3 hours) and RNNs (4 hours) on GTX 1080 Ti GPU.

### E.2  BLOCK DECODERS

### E.2.1  CNNS

**Architecture.** For block decoding using Convolutional Neural Networks (CNNs), we use a ResNet-like architecture (He et al., 2015), with the primary difference being the use of 1D convolutions instead

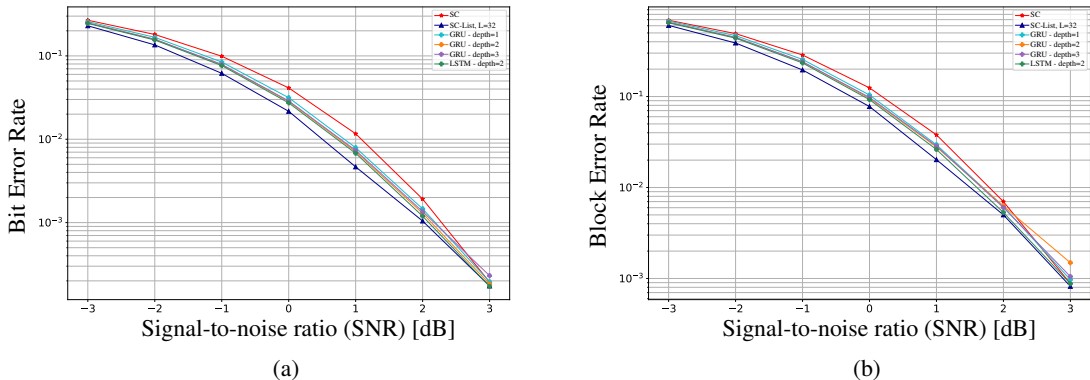

(a)            (b)

Figure 22: Polar$(64, 22)$: LSTMs and GRUs achieve similar reliability.

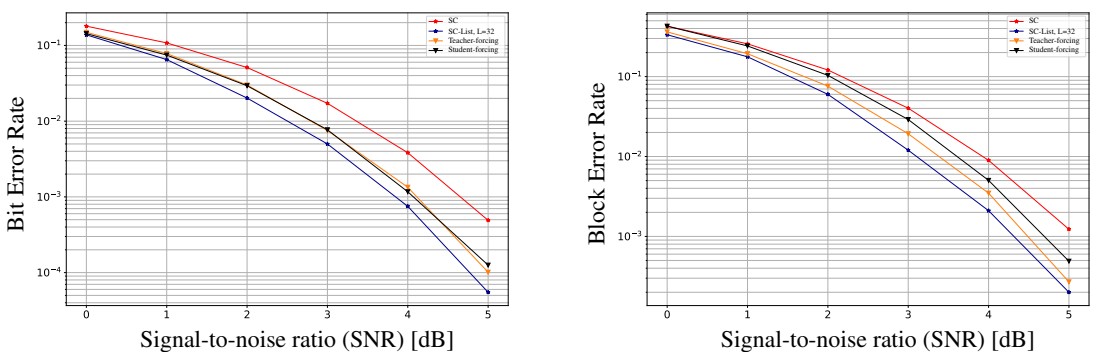

Figure 23: Training CRISP using student forcing results in sub-optimal BLER.

of 2D. The model has 10 1D-convolutional layers with residual connections skipping every two consecutive layers. Each convolutional layer has 64 channels, which are flattened at the penultimate layer and fed as an input to a fully-connected neural network with one hidden layer. We use the GELU (Hendrycks & Gimpel, 2016) activation function throughout the network. The model has 2.5M parameters for blocklength 64.

**Training.** We train the CNN model for $5,000$ iterations for each intermediate subcode of the curriculum. In the last step of the curriculum, we train it for $100,000$ iterations with a decaying cosine annealing schedule for the learning rate (Loshchilov & Hutter, 2016).

### E.2.2   BERT

**Architecture.** The model consists of 6 transformer blocks with unmasked self-attention and GELU activation. In each block, the multiheaded attention unit has 8 heads, and an embedding/hidden size of 64 is used throughout the network. The output vectors of the final transformer block are passed through a linear layer to estimate all the bits in one shot. The model has 350K parameters for blocklength 32.

**Training.** We train this model on each intermediate subcode for around $10,000 - 20,000$ steps. Thus the BERT-based decoder achieves better reliability than its GPT counterpart despite fewer training iterations (Fig. 19).

## F   RELIABILITY-COMPLEXITY COMPARISON

Two important metrics in evaluating a decoding algorithm are the decoding reliability and complexity. In this paper, we focus on optimizing the BER performance; the main goal of our paper is to design a curriculum based decoder for Polar and PAC codes that can achieve near-optimal reliability performance as opposed to the current data-driven approaches that only match the SC. In Sec. 4.2, we demonstrated that CRISP achieves excellent inference throughput on GPUs. We also see that the decoding complexity of CRISP can be improved with a hardware-aware neural architecture.

We believe that neural decoders, coupled with the recent advances in distillation Sanh et al. (2019) and pruning of neural networks Hinton et al. (2015); Wang et al. (2020); Anwar et al. (2015) far larger than ours (E.g., 110M for BERT vs. 2.5M for CRISP), can achieve even better runtimes. For instance, TinyBERT (Jiao et al. (2019)) uses knowledge distillation to learn a model 9.4x faster on inference compared to the parent BERT. With these improvements (a separate line of research), we can have a fair comparison with SCL decoders, which are the outcome of a decade of innovations in efficient implementations (e.g., FastSCL) – the CRISP decoder is only one of the first of its kind in designing a reliable neural decoder for Polar and Polar-like codes. Coupled with efficient GPU implementations, which are optimized for vector-matrix multiplications, and the aforementioned compression techniques, we believe neural decoders offer a great potential for fast and reliable channel decoding.

It is important to note that inference throughput is hardware and software dependant. In Table 1, we report throughput numbers of the optimized C++ multithreaded implementation of SC/SCL decoding on CPU using the aff3ct toolbox (Cassagne et al. (2019)). There has been progress in developing GPU implementations of SCL (Cammerer et al. (2017b); Han et al. (2017)). Since we could not find publicly available implementations of these works, we report throughput numbers of our implementation.

