# OpenReview forum: "CRISP: Curriculum based Sequential neural decoders for Polar code family"
_ICLR.cc/2023/Conference — Submitted to ICLR 2023_

### Official Review · Reviewer_1zGS · 2022-10-29

**Confidence:** 4
**Correctness:** 3
**Technical Novelty And Significance:** 2
**Empirical Novelty And Significance:** 1
**Recommendation:** 5

**Clarity, Quality, Novelty And Reproducibility:**

The proposed approach is described in a very clear way and seems to be novel, but it does not provide any benefits compared to state-of-the-art techniques used in coding theory.

**Strength And Weaknesses:**

The paper is very well written, and the proposed training method seems to be novel, although I am not an expert in machine learning.

The main problem with the paper is that it lacks proper complexity comparison with state-of-the-art decoding techniques for polar codes, and throughput results presented in Table 1 are quite disappointing. In particular, this table implies that the authors obtained decoder throughput for (32,16) code 13008 b/s for SCL(L=4) and 533333 b/s for CRISP.  However, it was shown in  [L2019] that SCL decoder with larger list size L=8 for much longer code (128,64) provides single-threaded throughput more that 3.5 Mb/s.  The discussion after Table 1 suggests, that the obtained results correspond to a multi-threaded implementation, so actual throughput gap is even worse.

Please observe that the SCL decoder is not the best one in terms of throughput.  Much more efficient decoding algorithms are known, e.g. FastSCL, sequential and trellis-based decoding algorithms [A2019,T2018,T2020]. The authors do not provide performance and complexity comparison with these techniques.  Lack of comparison with sequential decoders is particularly disappointing, since the authors themselves introduced a sequential-type decoder.

The authors use throughput of a software decoder as a complexity benchmark. This approach is quite unreliable and difficult to interpret, since it depends on software engineering skills and hardware platform features.  The reviewer suggests to use the number of arithmetic operations performed by the decoder as a benchmark. This would enable comparison with other published results.

The authors use non-standard notation (k,n) to denote the parameters of polar codes. The standard coding theory notation is (n,k), where n is code length and k is code dimension.

[L2019] Léonardon, M., Cassagne, A., Leroux, C. et al. Fast and Flexible Software Polar List Decoders. J Sign Process Syst 91, 937–952 (2019).
[A2019] M. H. Ardakani, M. Hanif, M. Ardakani and C. Tellambura, "Fast Successive-Cancellation-Based Decoders of Polar Codes," in IEEE Transactions on Communications, vol. 67, no. 7, pp. 4562-4574, July 2019
[T2018] P. Trifonov, "A Score Function for Sequential Decoding of Polar Codes," 2018 IEEE International Symposium on Information Theory (ISIT), 2018, pp. 1470-1474
[T2020] P. Trifonov, "Recursive Trellis Decoding Techniques of Polar Codes," 2020 IEEE International Symposium on Information Theory (ISIT), 2020, pp. 407-412

**Summary Of The Paper:**

The authors propose a decoding method for polar-like codes. The proposed approach is based on machine learning techniques, and novel training approach introduced by the authors. The authors show that the proposed method provides performance close to that of the successive cancellation list decoder.

**Summary Of The Review:**

The paper is well-written, but the paper lacks proper comparison, and does not seem to provide any improvement with respect to state of the art.

---

> ### Author Response · Authors · 2022-11-15
> **Throughput comparison**
>
> Thank you for the constructive feedback and insightful comments. We would like to emphasize that the main goal of our paper is to design a curriculum based decoder for Polar and PAC codes that can achieve near-optimal reliability performance as opposed to the current data-driven approaches that only match the SC. We also demonstrate that our curriculum based training achieves significant reliability gains on other neural architectures, such as 1D-CNNs (Figure 14a). In view of this, we had not tried to fully optimize the decoding complexity of CRISP (our view is that designing efficient circuit and software implementations is a serious and separate research direction). Having said that, we did conduct experiments to optimize our implementation complexity – we had not emphasized these experiments in the main part of the paper (because we did not consider these optimizations as part of our core contributions) and it is possible they were missed (Appendix C). Our preliminary experiments already provide encouraging evidence of optimization without sacrificing performance - the existing CRISP-CNN model (Appendix C.2) obtains a throughput of ~11.4Mb/s (0.0014s to decode a batch of 1000 samples of Polar(16,32)) on a single GTX 1080 Ti GPU.
>
> We posit that  further improvement in throughput can be realized using techniques like pruning and knowledge distillation [1-6]. These methods have been successful in speeding up neural networks far larger than ours (110M for BERT vs. 2.5M for CRISP). For instance, TinyBERT [1] uses knowledge distillation to learn a model 9.4x faster on inference compared to the parent BERT. With these improvements (a separate line of research), we can have a fair comparison with SCL decoders, which are the outcome of a decade of innovations in efficient implementations (e.g., FastSCL) – the CRISP decoder is only one of the first of its kind in designing a reliable neural decoder for Polar and Polar-like codes. $$\newline$$
> About the complexity metric: it is true that throughput is hardware and software dependent. However, it is also true that the total number of arithmetic operations (FLOPs) does not fully reflect the real runtime since it does not take into account the degree of parallelism of the algorithms [7, 8]. It is known in the literature that two models having similar FLOPs can have significantly different speeds depending on the platform and the parallelization (Section 1 of [8] and references therein). Neural decoders capitalize on the inherent parallelization of GPUs which are heavily optimized for the matrix-vector multiplications. Coupled with these efficient GPU implementations and aforementioned compression techniques, we believe neural decoders offer a great potential for fast and reliable channel decoding.
>
>  We have updated the complexity comparison with throughput numbers of non-neural decoders whose code is publicly available. Specifically, we use an optimized C++ multithreaded implementation of [L2019] using the aff3ct toolbox [9].  We have revised the paper to better reflect our claims and included a corresponding detailed discussion about the complexity (Section 4.2 and Appendix F).
>
> We used the (k,n) notation since the closely-related Reed-Muller codes use the (r,m) notation. We have updated this; we thank the reviewer for bringing this to our notice.
>
> $$\newline$$
> [1] Jiao, Xiaoqi, et al. "Tinybert: Distilling bert for natural language understanding." arXiv preprint arXiv:1909.10351 (2019).
>
> [2] Sanh, Victor, et al. "DistilBERT, a distilled version of BERT: smaller, faster, cheaper and lighter." arXiv preprint arXiv:1910.01108 (2019).
>
> [3] Hinton, Geoffrey, Oriol Vinyals, and Jeff Dean. "Distilling the knowledge in a neural network." arXiv preprint arXiv:1503.02531 (2015)
>
> [4] Cheng, Yu, et al. "Model compression and acceleration for deep neural networks: The principles, progress, and challenges." IEEE Signal Processing Magazine 35.1 (2018): 126-136.
>
> [5] Wang, Ziheng, Jeremy Wohlwend, and Tao Lei. "Structured pruning of large language models." arXiv preprint  (2019).
>
> [6] Anwar, Sajid, Kyuyeon Hwang, and Wonyong Sung. "Structured pruning of deep convolutional neural networks." ACM Journal on Emerging Technologies in Computing Systems (JETC) 13.3 (2017): 1-18.
>
> [7] Idelbayev, Yerlan, and Miguel Á. Carreira-Perpiñán. "Beyond FLOPs in low-rank compression of neural networks: Optimizing device-specific inference runtime." 2021 IEEE International Conference on Image Processing (ICIP). IEEE, 2021.
>
> [8] Ma, Ningning, et al. "Shufflenet v2: Practical guidelines for efficient cnn architecture design." Proceedings of the European conference on computer vision (ECCV). 2018.
>
> [9] Cassagne, Adrien, et al. "Aff3ct: A fast forward error correction toolbox!." SoftwareX 10 (2019): 100345.

---

> > ### Comment · Reviewer_1zGS · 2022-11-16
> > **Comments on revised paper**
> >
> > Unfortunately, the paper still lacks performance and complexity comparison with state-of-the-art sequential decoding algorithms for polar codes.
> >
> > Decoding throughput obtained by the authors for the GPU-based implementation is much worse compared to the published results for classical decoders.  The best throughput obtained by the authors is 15.72 Mbps, while the papers listed below report decoding throughput 41-1600 Mbps for much longer codes.
> >
> > Hence, I believe that the approach considered in the paper has little practical merit for the coding community.  However, it may be interesting from the point of view of machine learning.
> >
> > X. Han, R. Liu, Z. Liu and L. Zhao, "Successive-cancellation list decoder of polar codes based on GPU," 2017 3rd IEEE International Conference on Computer and Communications (ICCC), 2017, pp. 2065-2070
> > Z. Liu, R. Liu, Z. Yan and L. Zhao, "GPU-based Implementation of Belief Propagation Decoding for Polar Codes," ICASSP 2019 - 2019 IEEE International Conference on Acoustics, Speech and Signal Processing (ICASSP), 2019, pp. 1513-1517

---

> > > ### Author Response · Authors · 2022-11-17
> > > **Throughput on GPU after optimization**
> > >
> > > Thank you for the quick response.
> > >
> > > By choosing the optimum batch size for efficient inference on GPUs, we now obtain a throughput of  $\textbf{133Mbps}$ for Polar(64, 22) and $\textbf{250Mbps}$ for Polar(32,16) codes. We have updated Tables 1 and 2 accordingly. These throughput numbers are comparable to the throughput of $\textbf{71Mbps}$ obtained by the cited work [H2017] on Polar(512, 256) for L=32 using an efficient GPU implementation. We highlight that these improvements were obtained by just optimizing overnight (past 24 hours) and further improvement in our inference throughput is possible using techniques mentioned in the previous comment (again, demonstrating high computational performance was not the focus of this work).
> > >
> > > Separately, we could not find publicly-available implementations of the sequential decoding algorithms [H2017, T2018, A2019], and the BP decoding algorithm [L2019B] (which is only marginally better than SC) cited by the reviewer. Further the results in these works are for block lengths greater than 256 which is not the focus of our paper. Hence we did not directly compare CRISP to these works in Table 1. However, we have added comparisons to [L2019], which we independently verified, in the revised paper.
> > >
> > > Notes: We used GTX 1080-Ti GPUs in our experiment, comparable to the TitanX-GPUs used in [H2017].
> > >
> > > — - - - - - - - - - - - - - - - - - – - - - - - – - - - - - – - - - - - - – - - - - – - - - - - - - – –  – - – - - - – - - - –  - - - - - - - - - - - - - - - - - - - -  - - - - --
> > >
> > > Throughput calculation details:
> > > In the earlier version of the paper, we measured the throughput by computing the amount of time required to decode a batch of 1000 samples. However, the inference throughput on a GPU can be optimized by choosing efficient parameters; one of them being the batch size.
> > > For Polar(64, 22) using a batch size = 10000, we were able to achieve a throughput of 133Mbps; and 250Mbps for Polar(32, 16) using a batch size of 25000 for the CRISP_CNN decoder.
> > >
> > >
> > > [T2018] P. Trifonov, "A Score Function for Sequential Decoding of Polar Codes," 2018 IEEE International Symposium on Information Theory (ISIT), 2018, pp. 1470-1474
> > >
> > >
> > >
> > > [L2019] Léonardon, M., Cassagne, A., Leroux, C. et al. Fast and Flexible Software Polar List Decoders. J Sign Process Syst 91, 937–952 (2019).
> > >
> > >  [A2019] M. H. Ardakani, M. Hanif, M. Ardakani and C. Tellambura, "Fast Successive-Cancellation-Based Decoders of Polar Codes," in IEEE Transactions on Communications, vol. 67, no. 7, pp. 4562-4574, July 2019
> > >
> > > [H2017] X. Han, R. Liu, Z. Liu and L. Zhao, "Successive-cancellation list decoder of polar codes based on GPU," 2017 3rd IEEE International Conference on Computer and Communications (ICCC), 2017, pp. 2065-2070
> > >
> > > [L2019B] Z. Liu, R. Liu, Z. Yan and L. Zhao, "GPU-based Implementation of Belief Propagation Decoding for Polar Codes," ICASSP 2019 - 2019 IEEE International Conference on Acoustics, Speech and Signal Processing (ICASSP), 2019, pp. 1513-1517

---

### Official Review · Reviewer_JMRY · 2022-11-02

**Confidence:** 4
**Correctness:** 4
**Technical Novelty And Significance:** 3
**Empirical Novelty And Significance:** 3
**Recommendation:** 6

**Clarity, Quality, Novelty And Reproducibility:**

It is easy to follow and moderately novel.
I don't think the result is not reproducible.

**Strength And Weaknesses:**

Strength
- It's an interesting application of curriculum learning to me. The rationale behind why the authors introduced curriculum learning for their purposes totally makes sense (feedback mechanism to overcome the key drawback of successive cancellation). I liked the key idea behind the curriculum training of CRISP. I feel it aligns well with what is happening when decoding the Polar codes.
- It is shown from the evaluation results that CRISP achieves better error-correcting performances compared to the baselines.

**Summary Of The Paper:**

This paper proposes a curriculum-based sequential neural decoder for Polar codes. It is shown that CRISP, trained based on information-theoretic insights, outperforms the SC decoder achieving near-optimal error-correcting performance.

**Summary Of The Review:**

Please see `Strength` section.

---

> ### Author Response · Authors · 2022-11-15
> **Acknowledgement**
>
> Thank you for the encouraging comments.

---

### Official Review · Reviewer_FDX2 · 2022-11-03

**Confidence:** 4
**Correctness:** 3
**Technical Novelty And Significance:** 2
**Empirical Novelty And Significance:** 2
**Recommendation:** 6

**Clarity, Quality, Novelty And Reproducibility:**

=================================================================

[Technical Comments]

The Inference throughput studies in Tables 1 and 2 are a little misleading because one column is based on GPU and the other on CPU, and also it also depends on the decoder implementations, which require more detailed descriptions.

=================================================================


**Strength And Weaknesses:**

=================================================================

[Main Strengths]

The authors' extensive descriptions of the backgrounds, as well as the important implementation codes, are included in the anonymous repo, which is the paper's main strength.

=================================================================

[Main Weaknesses]

The study's main shortcoming is that, while the performance improvements are intriguing, they do not appear to have a major impact, similar to the many flavors of data-driven channel decoders. I encourage the authors to make additional improvements, notably for the experiments mentioned in Section 4, which demonstrate how the CRISP handles non-Gaussian channels (e.g., memory channels) well with nonlinear GRU designs as compared to standard model-driven decoders.

=================================================================


**Summary Of The Paper:**

=================================================================

[Summary]

This paper presents an inductively biased curriculum-learning based training technique that makes use of a nesting hierarchy of polar codes to improve BER results through experimentation.

=================================================================


**Summary Of The Review:**

The usage of polar code nesting hierarchy for curriculum-learning based training strategy is a simple yet intriguing notion, however, it does not appear to make a significant impact, similar to the many kinds of data-driven channel decoders.

---

> ### Author Response · Authors · 2022-11-15
> **Non-Gaussian channels and throughput comparison**
>
> Thank you for the constructive feedback. Regarding the comment that data-driven decoders have had little impact, we respectfully disagree. Perhaps this was missed during the review (possible because it is featured deep in Appendix D), but we have already conducted the kind of experiments that the reviewer suggests. For instance, we see the robustness of the CRISP decoder is robust to deviations from the AWGN channel, while attaining similar performance over SC on Fading and T-distribution channels (cf., Figures 16,17 of Appendix D). Note that this decoder was trained on AWGN but directly tested on Fading and T-distribution channels. This suggests that the CRISP decoder inherits the robustness inherent to nearest neighbor decoding even though this was not explicitly featured in the training – this intuition is further justified by our experiments showing that the typical error events of CRISP match that of the optimal nearest neighbor (MAP) decoder – see Figure 8(b) of Appendix C.2.
>
> Regarding the comparisons in Tables 1 and 2, we agree with the reviewer. As the actual running times of several decoders are hardware and software-implementation dependent, the comparison is tricky. We have accordingly revised our paper to clearly state the comparison points - a new section (Appendix F) is added; a description of the new experiments and corresponding comparisons is in our response to Reviewer 1zGS.

---

### Official Review · Reviewer_aVwC · 2022-11-03

**Confidence:** 2
**Correctness:** 4
**Technical Novelty And Significance:** 3
**Empirical Novelty And Significance:** Not applicable
**Recommendation:** 8

**Clarity, Quality, Novelty And Reproducibility:**

The paper is clear and the authors do a good job of getting the main points across. I do not know of similar algorithms, so the result seems novel.

**Strength And Weaknesses:**

I am not an expert in neural networks, so my assessment here is far from authoritative.  To me, the main strength is showing a method of decoding which has some intuition behind it. That is, the decoding is done sequentially, like in SC decoding of polar codes. Another strength is that the authors seem to have found a complexity/error rate point that other algorithms cannot match. A weakness of this method is that it does not seem to work to moderate length codes, and only works for short codes. To their credit, the authors state this outright.

**Summary Of The Paper:**

The paper introduces a method to train a neural network to decode polar codes and related codes (PAC, polar with CRC). The main trick is that the training is gradual: first train to decode the first information bit, assuming all other information bits are frozen to 0, then start from the previous point and train assuming all but the second information bit is frozen to zero, etc. The resulting decoder is better than SC and worse than SCL in terms of bit error rate, but better than SCL in terms of throughput.

**Summary Of The Review:**

The result seems to be a good fit for ICLR

---

> ### Author Response · Authors · 2022-11-15
> **Acknowledgement**
>
> Thank you for the encouraging and fruitful comments.

---

### Decision · Program_Chairs · 2023-01-20

**Decision:**

Reject

**Justification For Why Not Higher Score:**

As described above, the contributions of this paper would be regarded as minor with regard to both machine learning and coding theory perspectives.

**Justification For Why Not Lower Score:**

N/A

**Metareview: Summary, Strengths And Weaknesses:**

This paper proposes a curriculum-learning-based neural decoders for polar and related codes. The main strength of this paper is to elucidate utility of curriculum learning in decoding of polar-like codes. The main weakness of this paper is that it does not seem contributing much to either machine learning or coding theory community: Contributions to the machine learning community would be minor in view of the prior work by Lee et al. (2020), whose proposal includes use of long short-term memory (LSTM) models and a version of curriculum learning (Lee et al. called it transfer learning) employing the C2N scheme. Given Lee et al.'s work, the proposal in this paper would be regarded as incremental, as the latter replaces LSTMs with gated recurrent units (GRUs) and the C2N scheme with the N2C scheme. Contributions to the coding theory community have also been questioned by Reviewer 1zGS. Collecting these pieces together, I would judge that contributions of this paper is not significant enough.